# CDK6 protects epithelial ovarian cancer from platinum-induced death via FOXO3 regulation

Alessandra Dall'Acqua[1], Maura Sonego[1], Ilenia Pellizzari[1], Ilenia Pellarin[1], Vincenzo Canzonieri[2], Sara D'Andrea[1], Sara Benevol[1], Roberto Sorio[3], Giorgio Giorda[4], Daniela Califano[5], Marina Bagnoli[6], Loredana Militello[3], Delia Mezzanzanica[6], Gennaro Chiappetta[5], Joshua Armenia[1], Barbara Belletti[1], Monica Schiappacassi[1],* iD & Gustavo Baldassarre[1],** iD

## Abstract

Epithelial ovarian cancer (EOC) is an infrequent but highly lethal disease, almost invariably treated with platinum-based therapies. Improving the response to platinum represents a great challenge, since it could significantly impact on patient survival. Here, we report that silencing or pharmacological inhibition of CDK6 increases EOC cell sensitivity to platinum. We observed that, upon platinum treatment, CDK6 phosphorylated and stabilized the transcription factor FOXO3, eventually inducing ATR transcription. Blockage of this pathway resulted in EOC cell death, due to altered DNA damage response accompanied by increased apoptosis. These observations were recapitulated in EOC cell lines *in vitro*, in xenografts *in vivo*, and in primary tumor cells derived from platinum-treated patients. Consistently, high CDK6 and FOXO3 expression levels in primary EOC predict poor patient survival. Our data suggest that CDK6 represents an actionable target that can be exploited to improve platinum efficacy in EOC patients. As CDK4/6 inhibitors are successfully used in cancer patients, our findings can be immediately transferred to the clinic to improve the outcome of EOC patients.

**Keywords** ATR; CDK6; FOXO3; ovarian cancer; platinum sensitivity
**Subject Categories** Cancer; Pharmacology & Drug Discovery; Urogenital System

## Introduction

Epithelial ovarian cancer (EOC) is the fourth leading cause of death for cancer in women and is characterized by late diagnosis, when tumor has already spread throughout the abdominal cavity in ~75% of the cases. Standard care for these patients combines radical surgery with platinum-taxol chemotherapy. Development of a platinum-resistant disease is a frequent event in advanced EOC patients and predicts poor prognosis (Jayson *et al*, 2014).

Molecular and morphological analyses divide EOC in two main subgroups, characterized by different driver mutations and different prognoses (Shih & Kurman, 2004; Lim & Oliva, 2013; Jayson *et al*, 2014). The largest subgroup (approx 75% of all EOC) comprises high-grade serous, high-grade endometrioid, and undifferentiated EOC and is characterized by p53 gene mutations, genomic instability, DNA copy number alterations, and few other distinct and recurrent mutations (Cancer Genome Atlas Research Network, 2011; Jayson *et al*, 2014).

The emergence of chemo-resistant clones greatly hampers treatment efficacy, and actionable recurrent mutations in relapsed resistant high-grade EOC have not been identified (Patch *et al*, 2015; Schwarz *et al*, 2015).

New clinical evidences emerging from patients treated with specific targeted therapies (i.e., PARP inhibitors) in combination with platinum (Oza *et al*, 2015; Mirza *et al*, 2016) demonstrate that combining targeted agents and chemotherapy improves platinum efficacy and may thus represent a valid therapeutic approach in well-selected EOC patients (e.g., those carrying BRCA1/2 mutations).

BRCA1/2 genes are hub regulator of DNA repair pathways (Siddik, 2003; Kelland, 2007) which act in concert with cell cycle regulators to dictate proper DNA damage response (DDR). Activation of DDR is essential to repair platinum-induced DNA damages and prevents gross genetic abnormalities to be inherited by daughter cells (Siddik, 2003; Kelland, 2007; Branzei & Foiani, 2008; Johnson & Shapiro, 2010; Trovesi *et al*, 2013). The activation of DDR preserves cancer cells from platinum-induced cell death, and, therefore, an altered DDR could contribute to increase platinum activity.

1   Division of Molecular Oncology, CRO Aviano, IRCCS, National Cancer Institute, Aviano, Italy
2   Division of Pathology, CRO Aviano, IRCCS, National Cancer Institute, Aviano, Italy
3   Division of Medical Oncology C, CRO Aviano, IRCCS, National Cancer Institute, Aviano, Italy
4   Division of Gynecology-Oncology, CRO Aviano, IRCCS, National Cancer Institute, Aviano, Italy
5   Genomica Funzionale, Istituto Nazionale Tumori -IRCCS- Fondazione G Pascale, Naples, Italy
6   Molecular Therapies Unit, Department of Experimental Oncology and Molecular Medicine, Fondazione IRCCS Istituto Nazionale dei Tumori Milan, Milan, Italy
    *Corresponding author. Tel: +39 0434 659 759/661; Fax: +39 0434 659 428; E-mail: mschiappacassi@cro.it
    **Corresponding author. Tel: +39 0434 659 759/661; Fax: +39 0434 659 428; E-mail: gbaldassarre@cro.it

A central role in the coordination of DDR is exerted by ATM/ATR kinases that transmit the signal of DNA damage and impose cell cycle arrest by blocking the activity of critical cyclin-dependent kinases (CDKs) (Johnson & Shapiro, 2010; Maréchal & Zou, 2013). It is well established that CDKs play a central role in regulating DDR (Jazayeri *et al*, 2006; Branzei & Foiani, 2008; Yata & Esashi, 2009; Trovesi *et al*, 2013). However, several open questions remain on the precise role of the different CDKs in this process (Wohlbold & Fisher, 2009; Hydbring *et al*, 2016) and on their role, if any, in regulating the response to platinum in high-grade EOC.

## Results

### CDK6 silencing sensitizes ovarian cancer cells to platinum

Given their role in regulating DDR, we hypothesized that CDK activity could also be involved in platinum sensitivity in high-grade EOC cells. To this aim, we used carboplatinum (CBDCA) at suboptimal doses (i.e., lower than IC50) in MDAH-2774 EOC cell line (hereafter MDAH), deriving from high-grade endometrioid tumors, and transduced cells with shRNAs targeting all 23 human CDKs (Fig 1A, Appendix Fig S1A and Appendix Table S1; Malumbres *et al*, 2009). Silencing of CDK6 and CDK17 significantly increased platinum-induced cell death, while silencing of other CDKs did not significantly alter survival with respect to control (Appendix Fig S1B). All CDKs expressed in MDAH cells at detectable level were efficiently silenced by at least one shRNA (Appendix Fig S1C).

We focused on the role of CDK6 that most significantly increased platinum-induced death in MDAH cells, and it is a potential actionable target (Finn *et al*, 2015; Murphy & Dickler, 2015).

First, since CDK6 is highly homologous to and shares with CDK4 the binding with D type cyclins, we further tested the silencing of CDK4 and CDK6 and confirmed that only silencing of CDK6 increased platinum sensitivity of MDAH cells (Appendix Fig S1D). To exclude off-target effects, we used four different shRNAs targeting CDK6 and observed that the efficiency of CDK6 silencing paralleled the efficacy of CBDCA-induced cell death (Fig 1B and C). Synthetic lethality was observed using either cis- or carboplatinum (CDDP or CBDCA, hereafter referred to as platinum), both in MDAH and in SKOV3ip cells (Fig 1C; Appendix Fig S2A). The colony formation assay confirmed the dose–response assay and the potential contribution of CDK6 in long-term exposure to platinum (Fig 1D and Appendix Fig S2B).

CDK6 is primarily involved in the regulation of G1 to S phase transition of the cell cycle, by phosphorylating, in complex with D type cyclins, the RB1 tumor suppressor gene (Hydbring *et al*, 2016; Tigan *et al*, 2016). However, in MDAH cells, CDK6 silencing alone did not significantly alter cell growth (Fig 1E). Accordingly, RB1 phosphorylation and cell cycle distribution were not significantly modified but, after platinum treatment, an increased cleavage of PARP1 (marker of apoptosis) was observed (Fig 1F and G). It has been reported that CDK6 inhibition induces cell senescence via the regulation of FOXOM1, also in the absence of detectable levels of RB1 (Anders *et al*, 2011). We thus tested whether CDK6 knockdown led to an increase in senescence in MDAH cells, treated or not with platinum. In line with what reported, we observed a little, although significant, increase in the number of β-Gal positivity in CDK6 silenced cells; however, this was equally present in untreated and platinum-treated cells (Fig 1H). Similarly, CDK6 silencing did not increase the number of β-Gal-positive SKOV3ip cells (Appendix Fig S2C and D), overall indicating that CDK6 knockdown marginally affects cell cycle progression and senescence in EOC cells and that these effects did not likely represent the cause of increased platinum sensitivity.

Conversely, we observed that MDAH cells silenced for CDK6 and treated with platinum displayed a pronounced induction of

**Figure 1. CDK6 silencing sensitizes EOC cells to platinum-induced cell death.**

A   Experimental design of the loss-of-function screening. "*n*" indicates the number of biological replicates of the screening (each performed in triplicate). Significance (*P*) was calculated by two-sided, unpaired *t*-test.

B   Cell viability of MDAH cells transduced with the indicated shRNAs and treated with 140 μg/ml of CBDCA for 16 h. The corresponding cell lysates were analyzed by Western blot. Data represent the mean ± SD of two independent experiments each performed in duplicate (two-sided, unpaired *t*-test).

C   CDDP and CBDCA dose–response curve of MDAH cells, silenced or not for CDK6. Results are expressed as percentage of viable cells with respect to untreated cells and the resulted IC50 (half maximal inhibitory concentrations) as the mean value of three independent experiments ± SD each performed in triplicate.

D   Colony assay and its quantification performed on MDAH cells transduced as indicated and treated or not with CDDP (0.3 μg/ml) for 72 h and then released for 72 h. Data represent the mean ± SD of two independent experiments performed in triplicate (two-sided, unpaired *t*-test).

E   Growth curve of control or CDK6 stably silenced MDAH cells. The corresponding cell lysate was analyzed by Western blot. Data represent the mean ± SD of two independent experiments (each performed in triplicate).

F   Western blot analysis of PARP1 cleavage and RB1, pRB1$^{S780}$, and CDK6 expressions in MDAH cells transduced as indicated and treated with vehicle (V) or with CBDCA, as in (B), and then released (R) for 8 and 24 h.

G   Cell cycle distribution of MDAH cells transduced with control or CDK6-specific shRNAs, evaluated by flow cytometry 72 h post-transduction. The corresponding cell lysates were analyzed by Western blot.

H   SA-β-Galactosidase-positive cells/field in MDAH cells transduced as indicated and stained 72 and 96 h post-transduction (left) or silenced and then treated with CBDCA, as in (B) (right). Data represent the mean ± SD of two independent experiments, each performed in triplicate (two-sided, unpaired *t*-test). Typical fields of stained cells are shown in the inset. Arrows point to β-galactosidase-positive cells.

I   Western blot analysis of the expression and cleavage of PARP1 and caspase-3 and the expression of γH2AX and CDK6, in MDAH cells silenced using control or CDK6-specific shRNAs and treated with vehicle (V) or with CBDCA as in (B) (P) and released (R) for the indicated times.

J   Immunofluorescence analysis evaluating the expression of γH2AX (red) in MDAH cells treated as indicated in (I). Nuclei were stained with propidium iodide (pseudocolored in blue). Release indicates cells treated with CBDCA as in (B) and allowed to repair for 8 h. Vinculin and tubulin were used as loading controls.

Data information: In each panel, significant differences are evidenced by asterisks (**$P < 0.01$, ***$P < 0.001$, ****$P < 0.0001$), and the exact *P*-values of (B, D and H) are reported in Appendix Table S4.
Source data are available online for this figure.

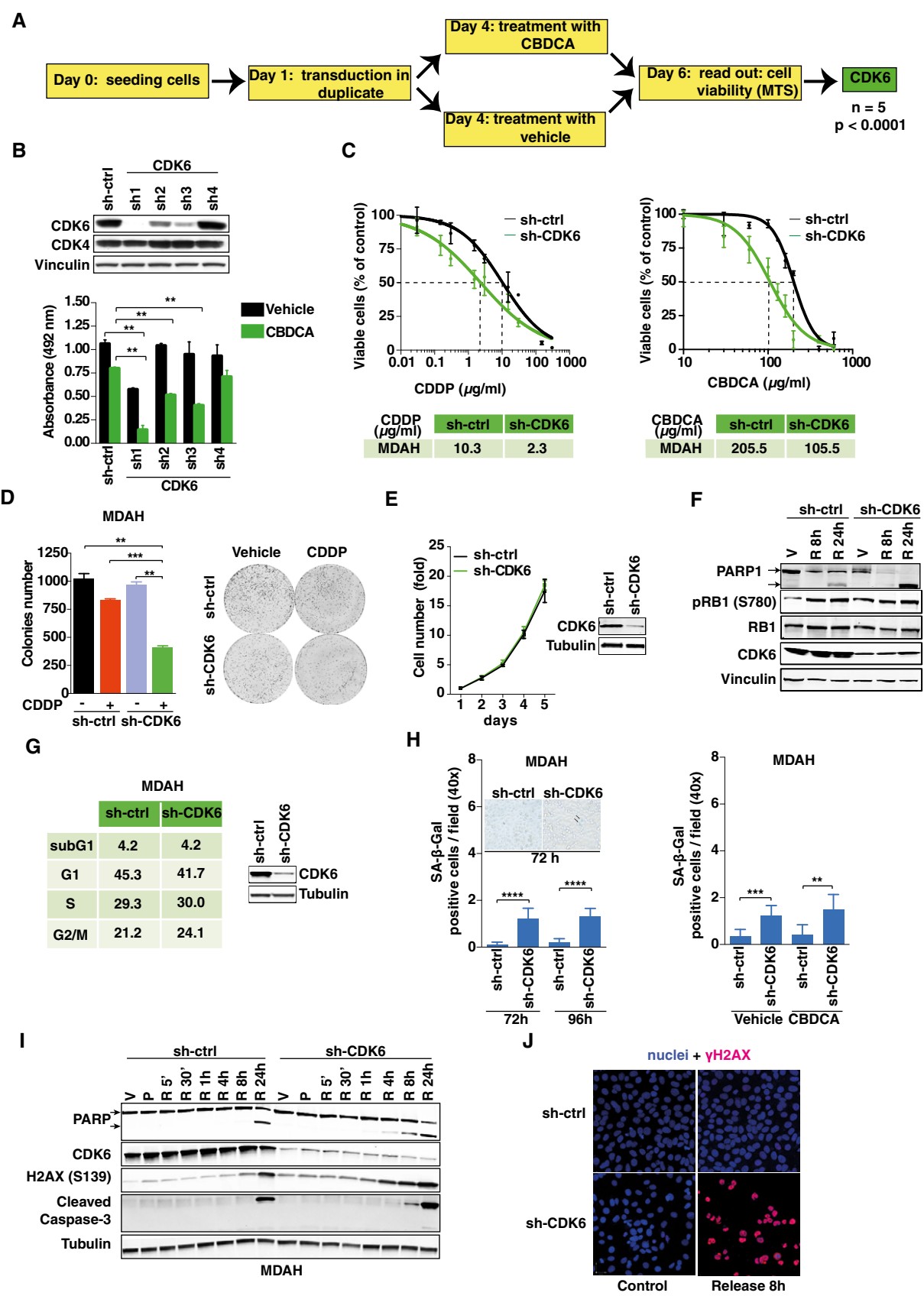

**Figure 1.**

apoptosis and an increased platinum-induced phosphorylation of histone H2AX$^{S139}$ ($\gamma$H2AX), used as early marker of DDR (Fig 1F and I). To confirm these results, we performed a time course analysis and evaluated the expression of $\gamma$H2AX, cleaved caspase-3, and cleaved PARP1, in cells treated with platinum and then allowed to repair for up to 24 h. These analyses, also confirmed by immunofluorescence staining, demonstrated that CDK6 knockdown resulted in higher and anticipated platinum-induced DNA damage, followed by increased apoptosis (Fig 1I and J).

We next sought to verify whether the control of platinum-induced DDR and apoptosis by CDK6 was linked to RB1 regulation. We took advantage of the RB1-null EOC cells OVCAR8 (Ha *et al*, 2000), and, as observed with MDAH and SKOV3ip cells, CDK6 knockdown in OVCAR8 resulted in increased platinum-induced death, with no alteration in proliferation and cell cycle distribution in the absence of platinum (Fig EV1A–C). We consistently observed in OVCAR8 cells that CDK6 knockdown resulted in higher and anticipated DNA damage and apoptosis, as demonstrated by the expression levels of $\gamma$H2AX and cleaved PARP1, respectively (Fig EV1D). OVCAR8 expressed low levels of endogenous CDK6 (Fig EV2A). We thus tested whether CDK6 overexpression could increase the resistance to platinum in this model and, accordingly, observed that CDK6-overexpressing cells were more resistant to platinum-induced cell death (Fig EV1E).

## CDK6 protects EOC cells from platinum via a kinase-dependent mechanism

CDK6 plays both kinase-dependent and kinase-independent roles (Kollmann *et al*, 2013; Hydbring *et al*, 2016; Tigan *et al*, 2016). To test whether the control of DNA damage and cell death in platinum-treated cells was kinase-dependent, we transfected wild-type (CDK6$^{WT}$), dominant-negative (CDK6$^{D163N}$), and constitutively active (CDK6$^{R31C}$) CDK6 vectors in OVCAR8 (RB1-negative, CDK6-low) and in MDAH cells (RB1-positive, CDK6-high). Expression of CDK6$^{D163N}$ reduced and CDK6$^{R31C}$ increased the survival of cells treated with low doses of platinum (Figs 2A and EV2A and B), suggesting that CDK6 kinase activity was necessary to protect cells from platinum. Next, we used the specific CDK4/6 inhibitor PD0332991 (Toogood *et al*, 2005) (hereafter referred to as PD) to test whether it recapitulated the results obtained with CDK6 silencing. As expected from the concomitant inhibition of CDK4 and CDK6, PD strongly decreased RB1$^{S780}$ phosphorylation in MDAH cells (Fig 2B) but had only minor effects on the response to platinum in KURAMOCHI and OVSAHO cells, displaying low/undetectable CDK6 levels (Fig EV2A, C, and D) or in MDAH cells stably silenced for CDK6 (Fig EV2E). Overall, these data support the possibility that PD activity on platinum-induced cell death was mainly CDK6-dependent. Interestingly, PD sensitized MDAH cells to platinum, especially when sequentially administered after platinum (Fig 2B).

To understand whether treatment with platinum and PD had a synergistic, additive, or antagonistic effect, we calculated the combination index (CI) of the two drugs, using platinum and PD at their IC25, IC50, and IC75, in MDAH, SKOV3ip, and OVCAR8 cells. In all cell lines and in all combination tested, the two drugs displayed a synergistic effect (CI < 1) that was particularly strong in SKOV3ip

cells, which are considered a model of platinum-resistant high-grade EOC cells (Fig 2C). This finding prompted us to test whether the platinum + PD combination could be particularly active in platinum-resistant cells. To this end, we generated MDAH platinum-resistant cells by exposing parental MDAH cells to 20 cycles of platinum treatment, using a 10-fold higher concentration of platinum of their IC50 and then waiting for their recovery (Fig EV2F). To mimic the *in vivo* situation, in which resistant clones coexist with a bulk sensitive population (Schwarz *et al*, 2015), we co-cultured resistant and parental cells at different ratios. Resistant clones were threefold more sensitive to platinum + PD than parental MDAH cells, and platinum + PD combination was more effective in preventing the emergence of platinum-resistant subclones with respect to platinum alone (Fig 2D–G).

Interestingly, MDAH platinum-resistant cells displayed higher expression of CDK6, both at mRNA and at protein level (Fig EV2G and H), suggesting a possible explanation of the increased synergism of PD with platinum observed in these cells.

Time-lapse microscopy confirmed that MDAH cells underwent massive cell death, particularly when the combination platinum + PD was used (Movies EV1, EV2, and EV3). The occurrence of cell blebbing and nuclear condensation and fragmentation supported the possibility that these cells underwent apoptosis (Movies EV4 and EV5).

To better investigate how PD sensitized cells to platinum, we first explored the possibility that senescence or alteration of cell cycle was involved in its synergistic activity with platinum. In line with what we previously observed in CDK6-silenced cells, PD increased the number of $\beta$-Gal-positive cells but their absolute number was only marginal and could not explain the massive cell death induced by the combination of platinum + PD (Fig EV2I). We next investigated whether PD could differently alter cell cycle progression when used alone or in combination with platinum. As expected from its ability to inhibit both CDK4 and CDK6 and, thereby, to decrease RB1 phosphorylation, PD induced a RB1-dependent cell cycle arrest in both MDAH and SKOV3ip cells, but not in RB1-null OVCAR8 (Figs 2H and EV2J). However, when used after platinum that induces a substantial S phase block, PD did not cause a G1 arrest but rather the accumulation of cells in S-G2 phases, followed by a fivefold to sevenfold increase of sub-G1 population with respect to platinum alone, indicating a strong increase in the apoptotic fraction (Figs 2H and EV2J).

## The CDK4/6 inhibitor PD0332991 increases platinum efficacy *in vivo*

We next moved *in vivo* and subcutaneously injected SKOV3ip or MDAH cells in nude mice, waited for tumor appearance (~50 mm$^3$), and then treated mice with vehicle, platinum, PD, or their combination (Figs 3A and EV3A). In line with the data collected *in vitro*, sequential administration of platinum and PD significantly reduced tumor growth, while the single treatments had minor effects (Figs 3A–C and EV3A–C). PD-treated mice displayed significantly reduced phosphorylation of RB1$^{S780}$ in tumors, both when administered alone or in combination with platinum, demonstrating that it was active *in vivo* at the dose used (Figs 3D and EV3D and E). Moreover, the platinum + PD combination was the most effective in increasing $\gamma$H2AX and in reducing cell proliferation (Ki67

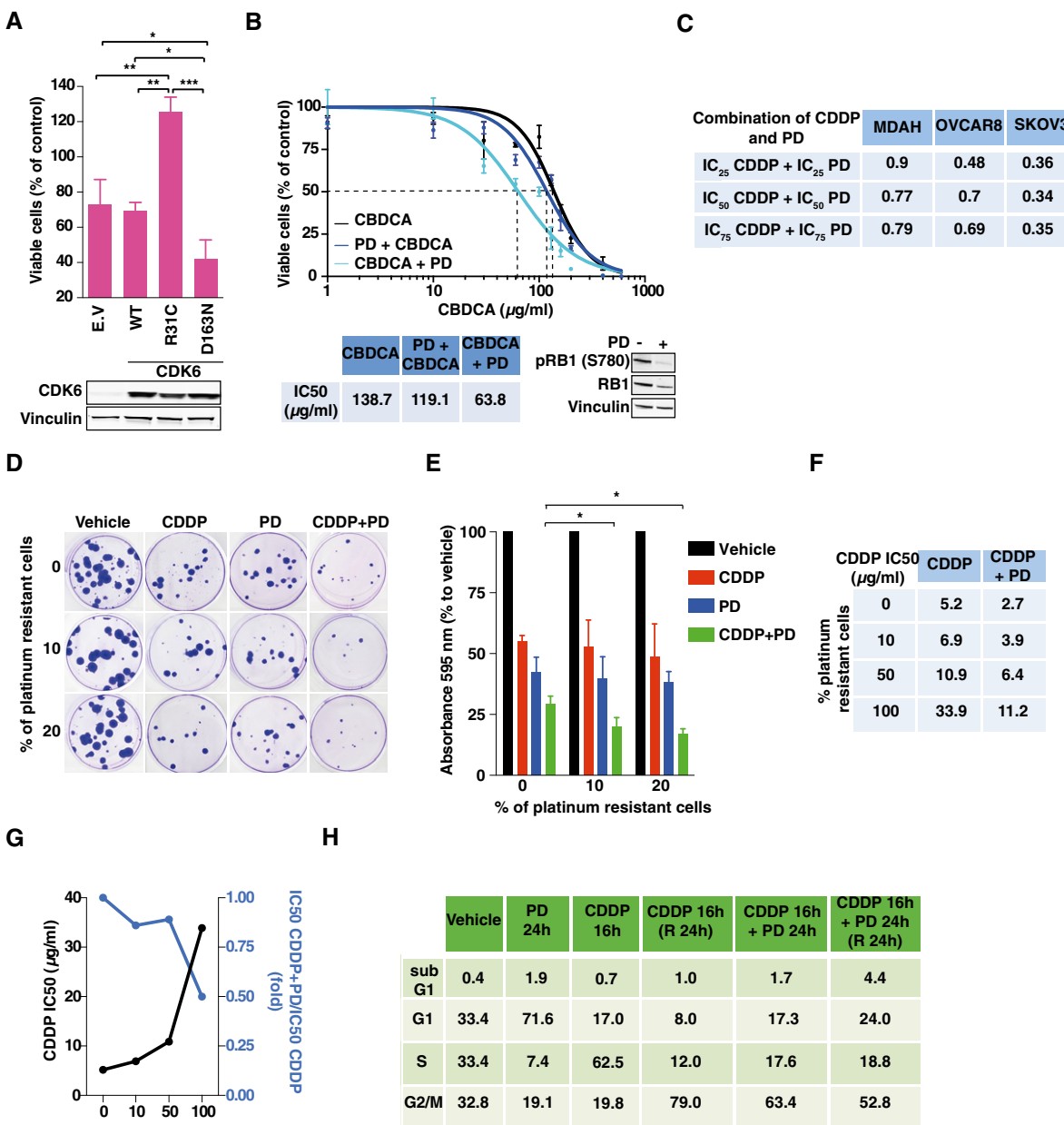

**Figure 2.  Inhibition of CDK6 kinase activity sensitizes EOC cells to platinum *in vitro*.**

A    Cell viability of CBDCA-treated OVCAR8 cells transfected with CDK6 WT, constitutive active (R31C), dominant-negative (D163N) mutants, or empty vector (E.V.) and treated with 60 µg/ml of CBDCA for 16 h. CDK6 expression is reported in the Western blot. Data represent the mean ± SD of two independent experiments performed in triplicate (two-sided, unpaired *t*-test).

B    Calculation of CDDP IC50 of MDAH cells treated or not with 8 µM PD0332991 (PD) using different schedules, as indicated. Expression and phosphorylation of RB1 were used as control of PD activity. Data represent the mean ± SD of three biological replicates.

C    Combination index (CI) resulting from the dose–response analysis of CDDP and PD treatments, alone or in combination. Data, analyzed with Calcusyn software, were used to measure if the two drugs have a synergistic (CI < 1), additive (CI = 1) or antagonistic (CI > 1) effect.

D, E    Colony assay (D) and quantification (E), using MDAH platinum-resistant clones plated with parental cells at different ratios and treated as indicated (PD 8 µM and CDDP 1.5 µg/ml). Data represent the mean ± SD of two independent experiments performed in triplicate (two-sided, unpaired *t*-test).

F    CDDP IC50 of mixed population of parental/platinum-resistant MDAH cells treated with increasing concentration of CDDP alone or CDDP + PD (8 µM) (*n* = 3 biological replicates).

G    Graph representing the CDDP IC50 (black line) and the normalized survival of cells treated with CDDP + PD combination (blue line) in cells plated and treated as described in (F).

H    Cell cycle distribution evaluated by flow cytometry of MDAH cells treated as in (D).

Data information: In each panel, significant differences are evidenced by asterisks (*$P < 0.05$, **$P < 0.01$, ***$P < 0.001$) and the exact *P*-values of (A and E) are reported in Appendix Table S4.

Source data are available online for this figure.

expression) (Figs 3D–F and EV3D–G). We then tested whether the platinum + PD combination could reduce the growth of larger SKOV3ip tumors (> 150 mm³) in nude mice. Also in this model of advanced platinum-resistant tumors, the combination of platinum + PD was effective in reducing both the tumor volume and weight (Fig EV3H–K). Similar results were obtained when tumors were established using MDAH cells stably silenced for CDK6 (Fig 3G–I). As observed *in vitro*, CDK6 silencing did not affect RB1 phosphorylation but increased platinum-induced DNA damage, displayed by γH2AX phosphorylation.

Overall, these data demonstrate that pharmacological or genetic targeting of CDK6 potentiates the effects of platinum on EOC cells, both *in vitro* and *in vivo*.

## FOXO3 is a CDK6-specific target able to modulate the sensitivity of EOC to platinum

Data collected so far indicated that the kinase activity of CDK6, but not of CDK4, protected EOC cells from platinum-induced death. Sicinski and colleagues reported that CDK6-preferential phosphorylation targets exist (Anders *et al*, 2011). We selected from their panel nine CDK6 targets that were phosphorylated better than RB1 and only poorly by CDK4 (Fig EV4A). We then targeted each of these CDK6 substrates with shRNA and observed that only the silencing of the transcription factor FOXO3 (also known as FOXO3A) mimicked the effects of CDK6 silencing in platinum-treated cells (Figs 4A and B, and EV4B and C).

FOXO3 overexpression overcame the increased platinum-induced cell death in CDK6-silenced cells, and, on the other side, FOXO3 silencing did not further sensitize MDAH cells to platinum (Figs 4C and D, and EV4D), overall confirming that FOXO3 acted downstream of CDK6 in response to platinum.

## CDK6/cyclin D3 complex phosphorylates FOXO3 on S325

Co-immunoprecipitation experiments demonstrated that CDK6 and FOXO3 barely interacted in untreated cells but that platinum

treatment increased their binding (Fig 4E). The modulation of FOXO3 and CDK6 interaction by platinum was time- and dose-dependent, as demonstrated by co-IP analyses (Fig 4E). We also observed that cyclin D3 was the principal D type cyclin involved in CDK6-FOXO3 binding and in the regulation of platinum sensitivity (Figs 4E and EV2E and F). Conversely, we did not detect any association between FOXO3 and CDK4 and cyclin D1 and/or their involvement in the regulation of platinum sensitivity (Figs 4E and EV4E and F). *In vitro* kinase assays confirmed that recombinant CDK6/cyclin D3, but not CDK4/cyclin D1, complex phosphorylated FOXO3 recombinant protein, suggesting a direct association between FOXO3 and CDK6/cyclin D3 complex also in living cells (Figs 4F and EV5A). *In silico* analyses identified eight serine residues in FOXO3 that could serve as CDK6 phosphorylation sites (Fig EV5B and C). Using FOXO3 deletion mutants, we mapped the region phosphorylated by cyclin D3/CDK6 between amino acids 315–344 (Figs 4G and EV5C and D). As a further confirmation of the specificity and relevance of this interaction, we immunoprecipitated endogenous CDK6 from MDAH cells treated or not with platinum for up to 16 h and observed that the immunoprecipitated complex phosphorylated GST-FOXO3 much better than GST-RB1 and that this phosphorylation was increased by platinum treatment (Fig 4H). In the phosphorylated region, two serines (S325 and S344) are predicted CDK6 targets and fulfill the requirements for being effectively phosphorylated, since they are surface-exposed and located in an intrinsically disordered region (Fig EV5E). Point mutation experiments demonstrated that S325 was the serine preferentially phosphorylated by cyclin D3/CDK6, but not by CDK4/cyclin D1 (Figs 4G and EV5F–H).

## CDK6 regulates FOXO3 stability and localization in platinum-treated cells

We noticed that CDK6 knockdown was associated with a decreased expression of FOXO3 (Fig 4D). Therefore, we better studied if CDK6 could regulate FOXO3 expression and/or localization in platinum-treated cells. Using cycloheximide (CHX), we observed that CDK6 knockdown decreased FOXO3 stability, particularly in

**Figure 3. PD potentiates platinum activity *in vivo*.**

A Schematic design of *in vivo* experiments with SKOV3ip xenografts testing the efficacy of suboptimal doses of CBDCA (20 mg/kg) and PD (150 mg/kg) alone and in combination.

B Analysis of tumor growth in each experimental group described in (A). Red arrows indicate CBDCA treatments; blue arrows indicate PD treatment (two-sided, unpaired *t*-test). Error bars represent SD.

C Scattered dot plot reporting the weight of explanted tumors. Each dot represents one tumor. Bars indicate mean ± 95% CI (Mann–Whitney test). Representative image of explanted tumors is shown on the top.

D Western blot analysis of pRB1[S780] and Ki67 expression in tumors explanted from mice treated as described in (A). Numbers on the top indicate the analyzed tumors. GAPDH was used as loading control.

E Immunofluorescence analysis of γH2AX (red) in tumors explanted from mice treated as described in (A).

F Quantification of γH2AX in explanted tumors (*n* = 3/group in which at least three randomly selected fields were studied) treated as indicated and evaluated as mean fluorescence per cell. Bars indicate mean ± 95% CI (Mann–Whitney test).

G Schematic design of *in vivo* experiments with MDAH transduced with sh-ctrl (right flank) or sh-CDK6 (left flank) and then subcutaneously injected in nude mice (*n* = 6 mice for each group).

H Analysis of explanted tumor weight is reported in the scattered dot plot. Bars indicate mean ± 95% CI. Western blot reports the expression and phosphorylation of H2AX and RB1 in tumors explanted from a representative mouse (Mann–Whitney test).

I Quantification of RB1 (upper graph) and H2AX (lower graph) phosphorylation in tumors explanted from four different mice treated as described in (G). Vinculin was used as loading control.

Data information: In each panel, significant differences are evidenced by asterisks (*$P < 0.05$, **$P < 0.01$, ***$P < 0.001$, ****$P < 0.0001$) and the exact *P*-values of (C, F, and H) are reported in Appendix Table S4.
Source data are available online for this figure.

platinum-treated cells (Fig 5A) and that, following platinum treatment, FOXO3$^{WT}$ was more stable than the non-phosphorylatable mutant FOXO3$^{S325A}$ (Fig 5B). Moreover, the low expression of FOXO3$^{S325A}$ in platinum-treated cells was reversed by inhibition of

ubiquitin-mediated proteasomal degradation (Fig 5C). Accordingly, by quantifying GFP-FOXO3 half-life in single cells, we calculated that the mean fluorescence/cell of FOXO3$^{WT}$ slightly increased following platinum treatment, while the one of FOXO3$^{S325A}$

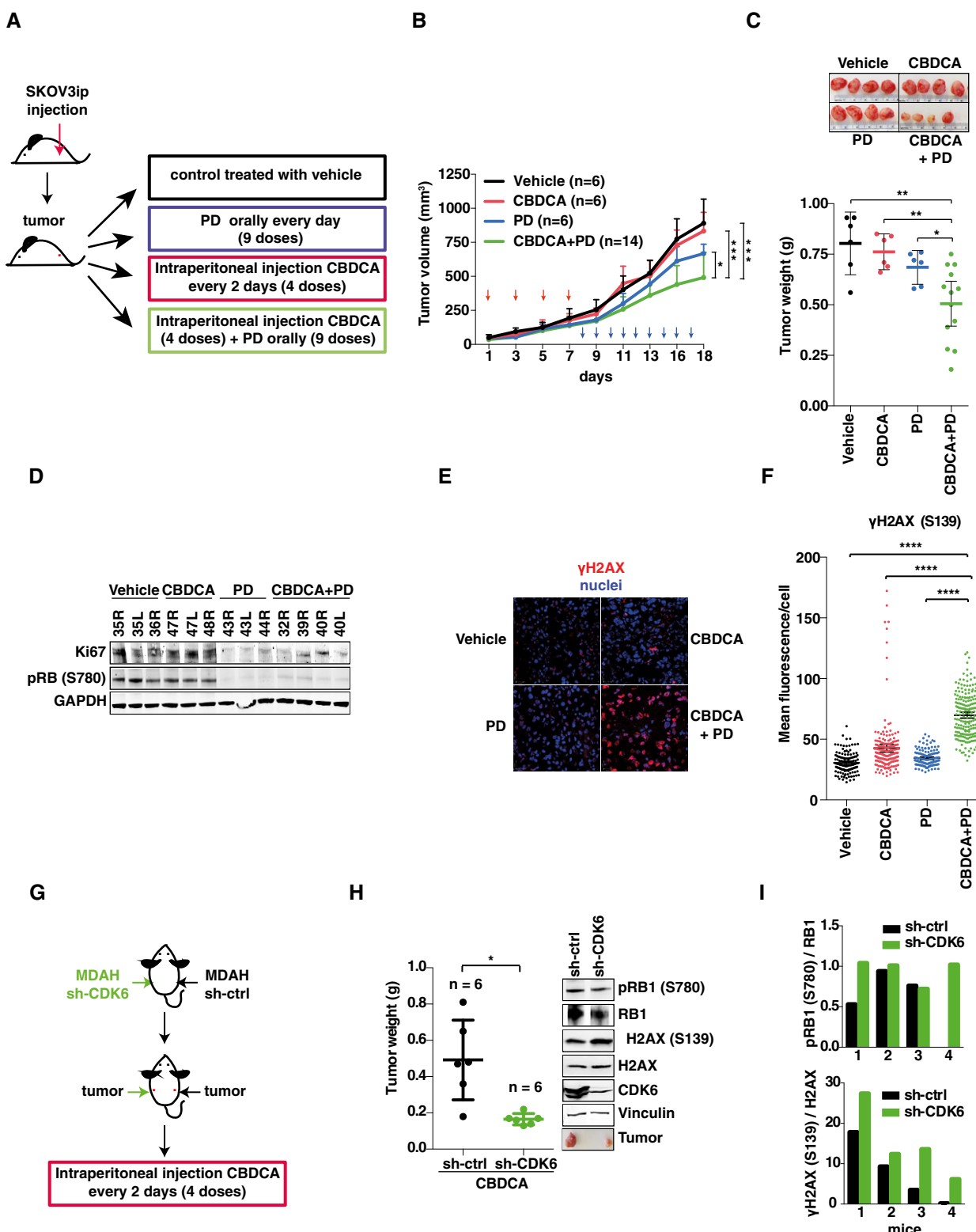

Figure 3.

significantly decreased (Fig 5D and E). Finally, FOXO3$^{WT}$, but not FOXO3$^{S325A}$, accumulated into the nucleus after platinum treatment (Fig 5D and F) and FOXO3$^{WT}$ overexpression, but not FOXO3$^{S325A}$, overcame the increased cell death induced by platinum in CDK6-silenced cells (Fig 5G).

We next tested whether the same findings could be recapitulated using PD treatment. To this aim, we differentially extracted nuclear and cytoplasmic proteins from cells treated with platinum, PD, or their combination and observed that FOXO3 nuclear expression increased by twofold upon platinum treatment and that this increase was completely prevented by PD treatment (Fig 5H). Furthermore, immunofluorescence analyses of tumors from mice treated with platinum in combination with PD (described in Fig 3) showed that the combination of platinum + PD reduced the total amount of FOXO3 and its nuclear localization also *in vivo* (Fig 5I).

### CDK6 controls platinum-induced cell death by regulating ATR via FOXO3

In high-grade EOC cells, response to platinum mainly relies on activation of ATM and ATR kinases (Siddik, 2003; Kelland, 2007; Maréchal & Zou, 2013). We observed that in CDK6- and FOXO3-silenced cells, ATR, but not ATM, expression was reduced and it could be rescued by concomitant FOXO3-overexpression in CDK6-silenced cells (Fig 6A and B). Accordingly, ATR expression was reduced in cells and in tumor xenografts, when treated with platinum + PD (Fig 6C and D). The ATR promoter contains three FOXO3 putative binding sites, identified *in silico* (Appendix Fig S3A). Chromatin immunoprecipitation (ChIP) experiments in cells transfected with FLAG-FOXO3 (Appendix Fig S3B) demonstrated that FOXO3 occupied the ATR promoter in a region comprised between −1,000 and −800, with respect to the transcription start site (TSS) (Fig 6E). Moreover, endogenous FOXO3 occupied the ATR promoter in the same region and this binding was increased under platinum treatment especially after the first 3 h of platinum release (Fig 6F and Appendix Fig S3C and D). Accordingly, we observed a twofold to threefold induction of ATR mRNA expression after platinum release

that was completely hampered by CDK6 or FOXO3 silencing (Appendix Fig S3E). Conversely, ChIP experiments using an anti-CDK6 antibody did not show any direct engagement of CDK6 to the ATR promoter, in any region and condition tested (Appendix Fig S3F), suggesting that CDK6 was not part of the transcriptional complex that regulated ATR expression following platinum treatment.

### Impairment of CDK6/FOXO3/ATR pathway induces PCC

It is well characterized that, in response to DNA damage, ATR phosphorylates and activates CHK1 which, in turn, blocks cell cycle progression and allows for proper DNA repair mechanisms to operate (Nghiem *et al*, 2001; Siddik, 2003). In the absence of either CDK6 or FOXO3, the activity of CHK1 was markedly reduced, as demonstrated by the decrease in CHK1$^{S296}$ phosphorylation (Fig 6A), observed as expected also in ATR-silenced cells (Fig 6G).

Our data showed that in EOC cells, platinum induced a prominent cell cycle block in S phase and that the subsequent treatment with PD increased the sub-G1 fraction, indicating an increase in apoptosis (Figs 2H and EV2J). It has been reported that ATR and CHK1 are both necessary to prevent the so-called premature chromosome condensation (PCC) a hallmark for mammalian cells that begin mitosis before completing DNA replication and that is invariably followed by cell death particularly when S phase is prolonged by pharmacological treatments, such as platinum, in the absence of a functional p53 (Nghiem *et al*, 2001).

We thus tested whether MDAH cells (harboring the p53$^{R273H}$ mutation) underwent PCC when silenced for CDK6, FOXO3, or ATR and treated with platinum. In control cells, platinum induced only a mild reduction of mitotic figures with respect to untreated cells, with no sign of PCC. In contrast, in cells silenced for CDK6, FOXO3, or ATR, more than 70% of total mitoses were categorized as PCC, following platinum treatment (Fig 6G–J). We also confirmed these data by metaphase spreads in control or CDK6-silenced cells, a technique that allows observing the morphology and integrity of condensed chromosome, which is lost in cells undergoing PCC (Appendix Fig S3G).

---

**Figure 4.   FOXO3 is a downstream target of CDK6.**

A    Viability of MDAH cells transduced with FOXO3 shRNAs and treated with CBDCA (140 μg/ml for 16 h). FOXO3 expression is reported. Data represent the mean ± SD of three independent experiments, each performed in triplicate (two-sided, unpaired *t*-test).

B    Calculation of CBDCA IC50 in FOXO3-silenced MDAH cells using two different FOXO3-shRNAs (three biological replicates). Error bars represent SD.

C    Upper panel: Time line of the experiment: day 1: MDAH cells were transfected with empty vector (E.V.) or FOXO3; day 2: transduction with control (ctrl) or CDK6 shRNAs; day 5: CDDP treatment (16 h); day 6: media change; day 7: cell viability assay. Lower panel: graph reports cell viability expressed as absorbance at 492 nm of cells transfected as indicated and treated with increasing concentration of CDDP. Data represent the mean ± SD of three biological replicates (two-sided, unpaired *t*-test).

D    Calculation of CDDP IC50 in MDAH cells transduced with control, CDK6, or FOXO3-specific shRNAs alone or in combination (three biological replicates). Expression of FOXO3 and CDK6 is reported in the upper inset.

E    Co-immunoprecipitation (IP) analysis of endogenous CDK6 in cells treated with vehicle (V) or with 7.5 or 15 μg/ml of CDDP for the indicated times. IPs were evaluated for the presence of CDK6, FOXO3, and cyclin D3, as indicated. In the lower panels, the expression of the same proteins in the corresponding lysates (Input) is reported. IgG indicates a lysate IP with unrelated antibody.

F, G    *In vitro* phosphorylation assay performed using recombinant cyclin D3-CDK6 complex and GST-RB1 fragment, FOXO3 full length as substrates (F), or the indicated FOXO3 deletion mutants carrying or not the S325A point mutation as indicated (G). C1: reaction mix plus recombinant kinase.

H    *In vitro* phosphorylation assay performed using CDK6 complex immunoprecipitated from MDAH cells treated with vehicle (V) or with CDDP 15 μg/ml for the indicated hours.

Data information: Tubulin, actin, or Ponceau staining were used as loading control, as indicated in each panel. In each panel, significant differences are evidenced by asterisks (*$P < 0.05$, **$P < 0.01$, ***$P < 0.001$) and the exact *P*-values of (A and C) are reported in Appendix Table S4.

Source data are available online for this figure.

## CDK6 expression levels predict EOC patient survival

To verify whether the collected data could have significance in the human pathology, we analyzed the expression of CDK6 in a panel of

primary HGEOC samples collected in our Institute (Appendix Table S2). CDK6 was generally well expressed (Fig 7A), and cyclin D3 was preferentially expressed among the D type cyclins (Appendix Fig S4A). High-grade EOC samples collected from patients

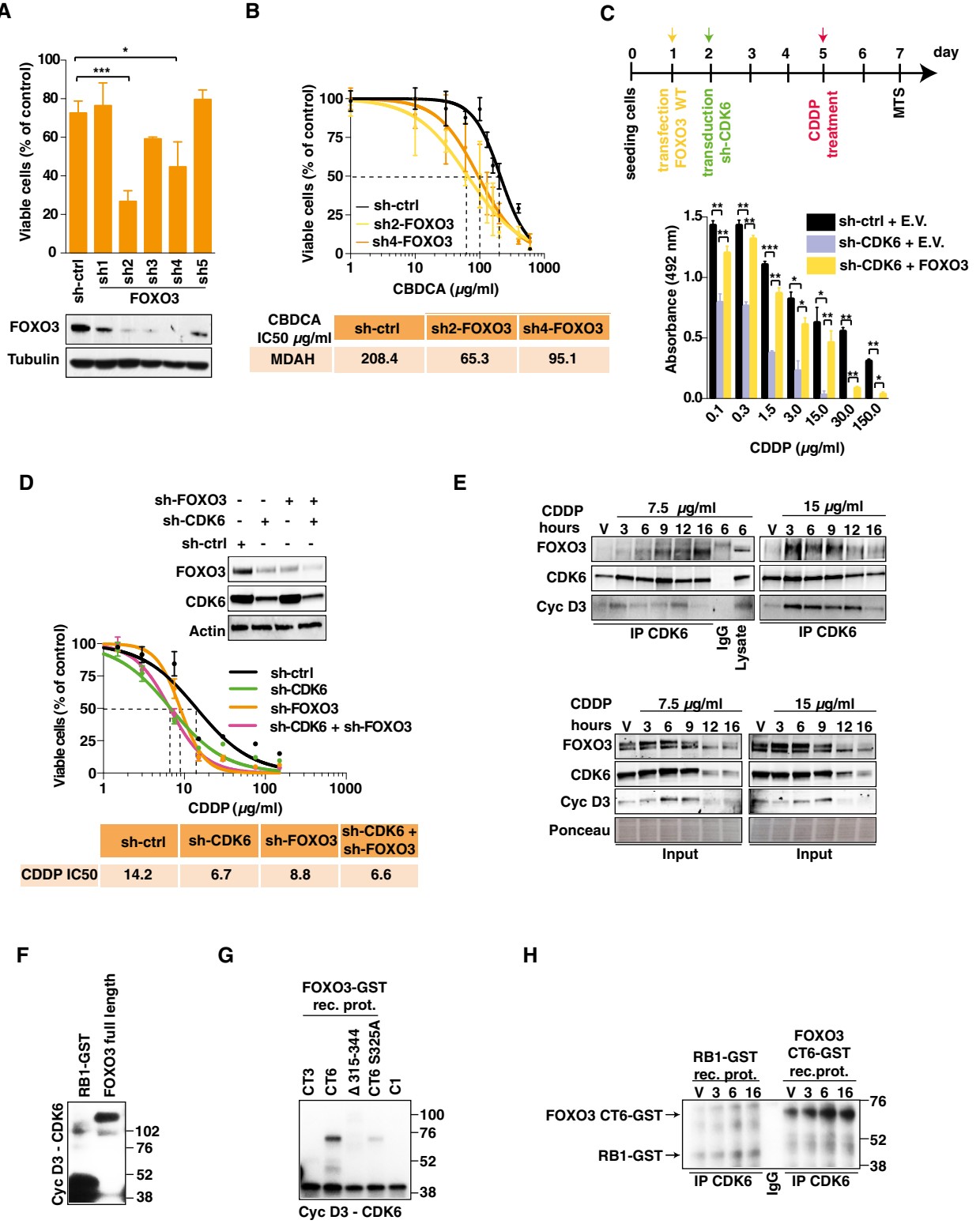

Figure 4.

who had received neoadjuvant platinum-based therapy or had a recurrent disease after previous platinum-based therapy displayed higher expression of cyclin D3 that readily co-precipitated with CDK6 (Fig 7B and Appendix Fig S4A). In the same tumors, the expression of CDK6, FOXO3, and ATR positively correlated (Appendix Fig S4B). We next tested the efficacy of platinum + PD in five primary high-grade EOC cultures that we established. Treatment with PD reduced of twofold to fourfold platinum IC50 in all but one cell culture (Fig 7C, upper panels and Appendix Fig S4C). Intriguingly, the "non-responder" primary culture was the only one derived from a naïve patient. The analysis of ATR, FOXO3, and CDK6 expression, in the same primary cultures treated with platinum and PD alone or in combination, confirmed that platinum induced an increased expression of ATR that was counteracted by the subsequent exposure to PD (Fig 7C, lower panels).

Finally, we sought to verify whether our findings correlated with patient survival, using the Tothill dataset (Tothill *et al*, 2008). Using univariate and multivariate analyses (including FIGO stage and suboptimal surgical debulking, which are the most relevant clinical prognostic factors for EOC patients), high CDK6 expression was significantly associated with early relapse [hazard ratio (HR) = 1.56, 95% confidence interval (CI) = 1.11–2.19, log-rank $P = 0.009$] (Fig 7D and Appendix Fig S4D). Similarly, using the KM Plotter online tool that include larger number of samples ($n = 1,307$), we observed that high CDK6 (but not CDK4) expression predicted a shorter progression-free survival (PFS) of EOC patients [hazard ratio (HR) = 1.45, 95% confidence interval (CI) = 1.17–1.79, log-rank $P = 0.00061$] and that the prognostic value of CDK6 was further reinforced if accompanied by high expression of ATR and FOXO3 [hazard ratio (HR) = 1.5, 95% confidence interval (CI) = 1.21–1.85, log-rank $P = 0.00016$] (Fig 7E and Appendix Fig S4E).

## Discussion

The survival of EOC patients mostly relies on the response to platinum-based therapy and the appearance of platinum-resistant

disease represents the worst prognostic factor for these patients (Jayson *et al*, 2014). Here, we show that CDK6 inhibition is synthetically lethal with platinum in EOC cells, describing a previously unknown pathway with potentially high clinical impact. We demonstrated that the platinum + PD combination was especially effective in preventing the re-growth of platinum-resistant cells and in inducing death in primary EOC cultures from platinum-treated patients. Since PD and other CDK4/6 inhibitors are in an advanced phase of clinical experimentation (Leonard *et al*, 2012; Dickson *et al*, 2013; Finn *et al*, 2015; Murphy & Dickler, 2015; Turner *et al*, 2015), our work contains relevant and potentially immediate translational implications. The feasibility of a maintenance therapy targeting the DDR after chemotherapy has been proved with PARP inhibitors in patients carrying BRCA1/2 germ line mutations and in platinum-sensitive EOC (Oza *et al*, 2015; Mirza *et al*, 2016). These clinical evidences suggest that test CDK4/6 inhibitor in similar settings is feasible, provided that accurate selection of patients that may benefit of this treatment is carried out. Based on our data, showing a particular efficacy in platinum-resistant EOC models, such as SKOV3ip and MDAH-resistant clones, we expect that EOC patients with acquired platinum resistance and high CDK6 and/or cyclin D3 expression will be the ones benefitting more from this association therapy.

It is important to note that CDK4/6 inhibitors, such as PD, are effective only in RB1-proficient tumors and are mainly cytostatic, due to the imposition of a cell cycle block in G1 phase (Leonard *et al*, 2012; Choi & Anders, 2014). As a consequence, it has been hypothesized that PD association with chemotherapy, which exerts its effect mainly on actively dividing cells, could be even detrimental (Roberts *et al*, 2012; Murphy & Dickler, 2015). Yet, our *in vitro* and *in vivo* data suggest that the sequential use of platinum followed by PD represents a treatment schedule that potentiates platinum efficacy by hampering the DDR induced by platinum. The same observation was also supported by others, showing that PD had antagonistic effects when administered

---

**Figure 5.  Phosphorylation by CDK6 regulates FOXO3 expression and localization in platinum-treated cells.**

A    Analysis of FOXO3 stability in control and CDK6-silenced MDAH cells treated with vehicle or CDDP 15 μg/ml for 3 h and then released in the presence of cycloheximide (CHX) for the indicated times. Densitometric quantification of the FOXO3 expression (mean ± SD) is reported ($n = 3$) (two-sided, unpaired $t$-test).

B    Analysis of FOXO3$^{WT}$ and FOXO3$^{S325A}$ stability in MDAH treated as in (A). Densitometric quantification of the FOXO3 expression (mean ± SD) is reported ($n = 3$) (two-sided, unpaired $t$-test).

C    Analysis of FOXO3$^{WT}$ and FOXO3$^{S325A}$ expression in cells treated as in (A) in the presence or not of MG132.

D    Expression and localization of GFP-FOXO3$^{WT}$ and GFP-FOXO3$^{S325A}$ (green) and γH2AX$^{S139}$ (red) in cells treated as indicated in (A). Phalloidin staining is pseudocolored in blue.

E, F    Quantification of FOXO3$^{WT}$ and FOXO3$^{S325A}$ expression (E) and localization (F) evaluated as mean fluorescence per cell (E) or folds over untreated cells (F), in cells treated as described in (D). V = vehicle; P = CDDP; R = 3 h release ($n = 2$ independent experiments in which at least six randomly selected field were analyzed). In (E), bars indicate mean ± 95% CI (Mann–Whitney test). In (F), bars indicate mean ± SD (Mann–Whitney test).

G    Viability of control or CDK6-silenced cells transfected with E.V., FOXO3$^{WT}$, and FOXO3$^{S325A}$ and then treated with CDDP as indicated. Data represent the mean ± SD (three biological replicates; two-sided, unpaired $t$-test). On the bottom, Western blot analyses show the expression of FOXO3$^{WT}$ and FOXO3$^{S325A}$ EGFP-tagged proteins and CDK6 in MDAH cells transfected and transduced as indicated.

H    Expression of FOXO3 in the nuclear (N) and cytoplasmic (C) fractions of MDAH cells treated or not with CDDP (15 μg/ml, 3 h) and PD (8 μM, 24 h), as indicated. The expression levels of RB1 and pRB1 (S780) are used to monitor PD activity. Graph reports the normalized quantification of nuclear fractions of FOXO3, obtained by densitometric scanning of the blot (FOXO3/PSF ratio).

I    Immunofluorescence analysis and quantification of FOXO3 expression in tumors explanted from mice ($n = 3$ per group in which at least three randomly selected fields were studied) (see Fig 3A) treated as indicated and evaluated as mean fluorescence per cell. Bars indicate mean ± 95% CI (Mann–Whitney test).

Data information: Tubulin, GRB2, and PSF were used as loading controls. In each panel, significant differences are evidenced by asterisks (*$P < 0.05$, **$P < 0.01$, ***$P < 0.001$, ****$P < 0.0001$) and the exact $P$-values of (A, B, E–G, and I) are reported in Appendix Table S4.
Source data are available online for this figure.

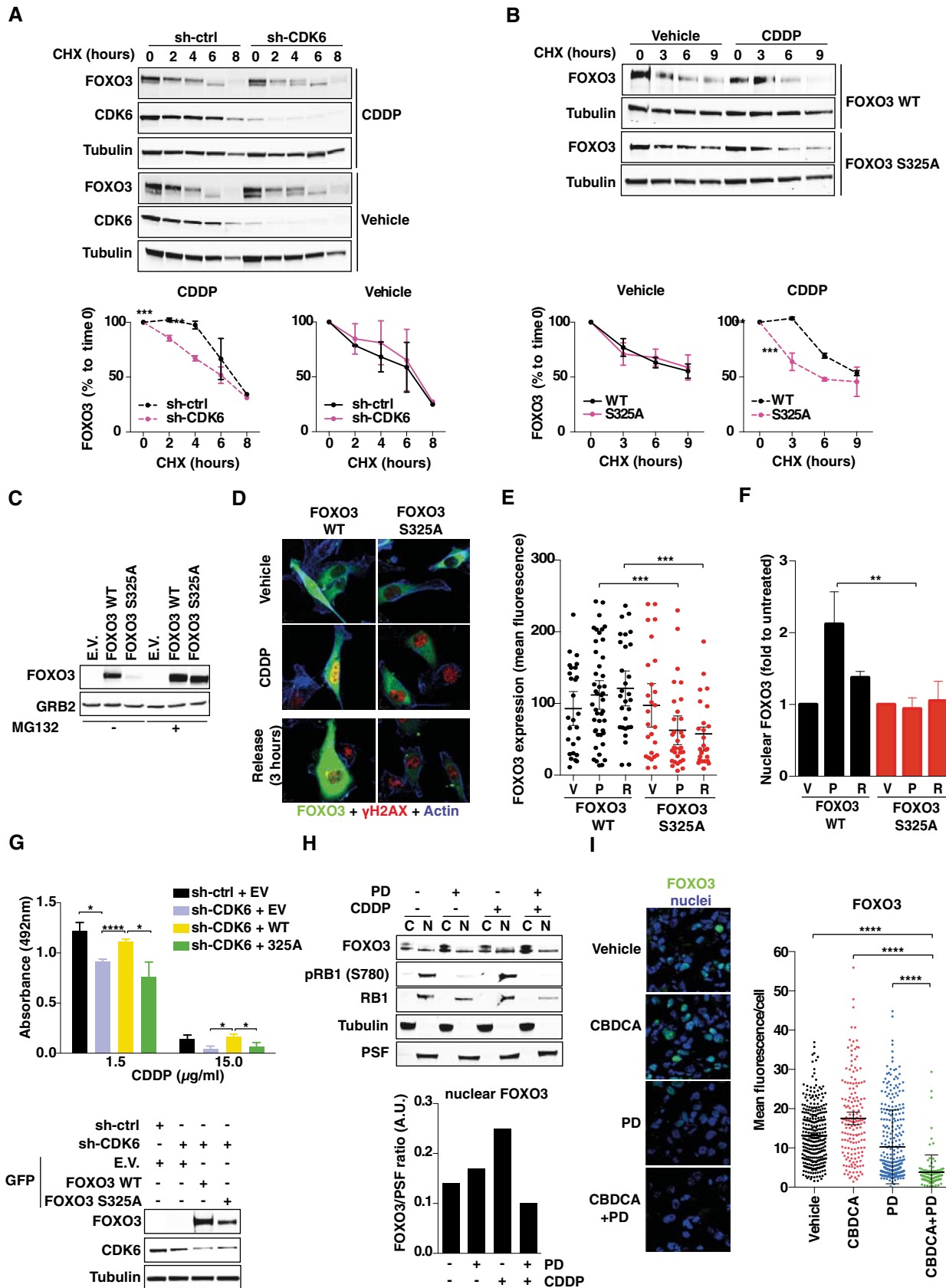

**Figure 5.**

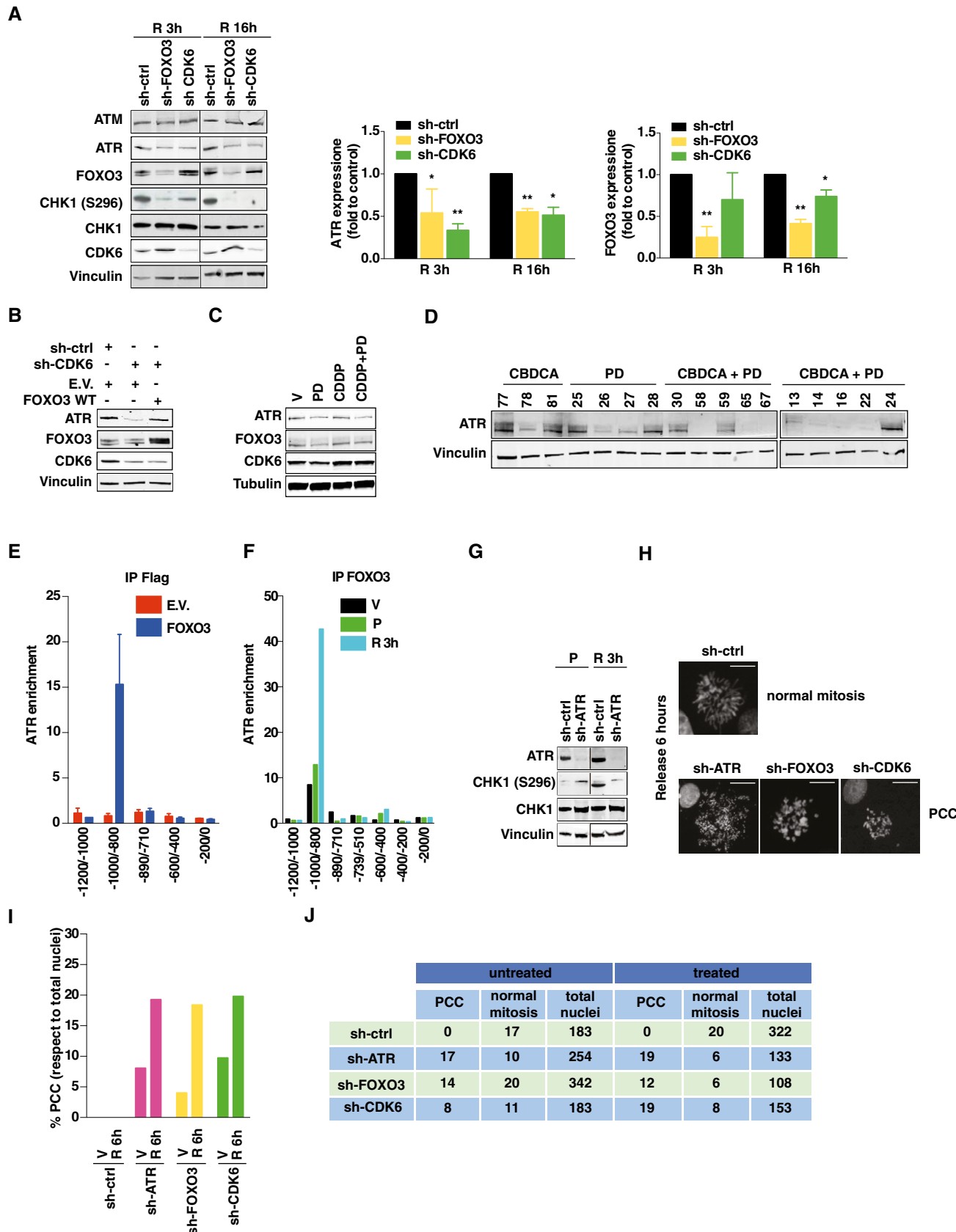

Figure 6.

**Figure 6.  CDK6 and FOXO3 regulate ATR expression and protect platinum-treated cells from premature chromosome condensation (PCC).**

A   Expression and phosphorylation of the indicated proteins, in MDAH cells silenced for CDK6 or FOXO3 treated with CDDP (15 μg/ml, 3 h) and released (R) for 3 or 16 h. Graph reports the normalized quantification of FOXO3 and CDK6, obtained by densitometric scanning of the blot (FOXO3/vinculin ratio). Data represent the mean ± SD of three independent experiments. Significance (*P*) was calculated by two-sided, unpaired *t*-test.

B   Expression of ATR in MDAH cells silenced for CDK6 and then transfected with FOXO3$^{WT}$, as indicated, and treated with CDDP as in (A).

C   Expression of ATR, FOXO3, and CDK6 in MDAH cells treated with CDDP (15 μg/ml, 3 h) and PD (8 μM, 24 h), alone or in combination.

D   ATR expression in tumors explanted from mice treated as indicated and described in Fig EV5A. The number on top indicates the analyzed tumors.

E   Enrichment of ATR promoter in FLAG-IPs in chromatin immunoprecipitation (ChIP) analysis using cells transfected with empty vector (E.V.) or FLAG-FOXO3. Data represent the mean ± SD of two independent experiments, each performed in duplicate. Data are expressed as fold enrichment with respect to control IgG.

F   Enrichment of ATR promoter in ChIP analysis, using an anti-FOXO3 or control-IgG antibodies in cells treated with vehicle (V) or with CDDP (15 μg/ml, 3 h) (P) and then released (R) for 3 h as in (A). Data represent the mean of three biological replicates. Data are expressed as fold enrichment with respect to control IgG.

G   Expression and phosphorylation of CHK1 in control and ATR-silenced MDAH cells treated with CDDP as in (F).

H   Typical images of normal mitosis and premature chromosome condensation (PCC), observed in MDAH cells transduced as indicated and then treated with CDDP and then allowed to recover for 6 h. Scale bar = 10 μm.

I   Percentage of PCC (with respect to total nuclei) in MDAH cells treated with CDDP (15 μg/ml, 3 h) and then released for 6 h (R 6 h) in platinum-free medium. V = vehicle.

J   Table reporting the number of normal mitosis, PCC, and total nuclei counted in cells treated as in (I).

Data information: Vinculin and tubulin were used as loading controls. In each panel, significant differences are evidenced by asterisks (*P < 0.05, **P < 0.01) and the exact *P*-values of (A) is reported in Appendix Table S4.

Source data are available online for this figure.

before, but not when administered after or with, platinum (Konecny *et al*, 2011). Indeed, our results suggest that in EOC, CDK6 is less involved in regulating G1-S transition and proliferation than in other models and its inhibition, in cells blocked in S phase by platinum, induced apoptosis, likely as a consequence of increased PCC. PCC is a molecular event amplified by the loss of functional p53 and in the absence of a functional ATR-CHK1 pathway (Nghiem *et al*, 2001). The fact that high-grade EOC almost invariably carry p53 mutations (Cancer Genome Atlas Research Network, 2011; Jayson *et al*, 2014) suggests that the association of chemotherapy and CDK6 inhibitors could be particularly effective in these tumors. Accordingly, the three cell lines and the five primary cultures we used in this study were all mutated for TP53. Although our data clearly indicate that inhibition of CDK6 increased EOC cell death by altering FOXO3-ATR expression, we cannot exclude that other pathways leading to apoptosis could also participate.

From a molecular point of view, we show that CDK6 binds and phosphorylates FOXO3, which, in turn, induces the expression of ATR. Although CDK4 and CDK6 can potentially bind the same D type cyclins to regulate G1-S phase progression (Choi & Anders, 2014), it is well established that each CDK and each D cyclin plays unique roles, both in normal development and during cancer progression (Malumbres *et al*, 2004; Malumbres, 2011; Choi *et al*, 2012; Kollmann *et al*, 2013).

Our data, showing that CDK6 phosphorylates and thereby stabilizes FOXO3, are highly reminiscent of the effects elicited on FOXM1, a related transcription factor, that, following phosphorylation by CDK4 and CDK6, is stabilized and transactivated to prevent senescence of cancer cells (Anders *et al*, 2011). Together, these results reinforce the notion that CDK4/6 are important regulators of the FOXO family of transcription factors. Interestingly, Foxo3 knockout in mice results in early depletion of functional ovarian follicles and in female sterility (Castrillon *et al*, 2003), indicating that FOXO3 plays a specific role in epithelial cells of the ovary.

Our data strongly indicate that FOXO3 is a key mediator of CDK6 activity, in response to platinum. However, we based our search for CDK6 substrates on a previous screening investigating only the potential nuclear targets of CDK4 and/or CDK6 (Anders *et al*, 2011). We thus cannot exclude that CDK6 has other relevant targets that could contribute to protect EOC cells from platinum-induced death that were not tested in this study. In the perspective of a clinical application, we are aware that our study has limitations inherent the low number of primary cultures tested and the predicted hematological toxicity of CDK4/6 inhibitors that could impact on the feasibility of their association with platinum in EOC patients. All these variables will need to be carefully taken into account when designing a clinical trial.

However, our work highlights a novel pathway that in EOC cells prevents platinum-induced cell death. This pathway is actionable and druggable and could be therefore exploited to improve response to platinum in recurrent EOC patients.

**Figure 7.  High CDK6 expression predicts poor prognosis in EOC patients.**

A   Percentage (left pie graph) and typical images of low (A) and high (B) CDK6 expression in a TMA containing 73 EOC samples.

B   Co-IP analysis of CDK6 with cyclin D1 and D3 in the indicated human EOC tumors. IgG = unrelated Ab. Lys = tumor cell lysates. Arrow points to the cyclin D3 band.

C   CDDP IC50 of primary cultures derived from the indicated tumors, in the presence or not of PD (three biological replicates ± SD). In the lower panels, the Western blot analyses evaluating the expression of ATR, FOXO3, and CDK6 in the same primary cultures, treated as indicated, is shown.

D   Kaplan–Meier survival curve evaluating the progression-free survival (PFS) in the Tothill dataset, based on the expression of CDK6 (upper panel). The lower table summarizes the significance of CDK6 in predicting PFS in univariate and multivariate analysis (Cox regression) using all malignant cases in the dataset (*n* = 218; events = 155; log-rank test was used to calculate significance). HR = hazard ratio; CI = 95% confidence interval.

E   Kaplan–Meier survival curve evaluating the significance of CDK6, CDK4, and the CDK6, FOXO3 and ATR combination, in predicting the PFS using the online tool KM Plotter (*n* = 1,307; events = 484; log-rank test).

Source data are available online for this figure.

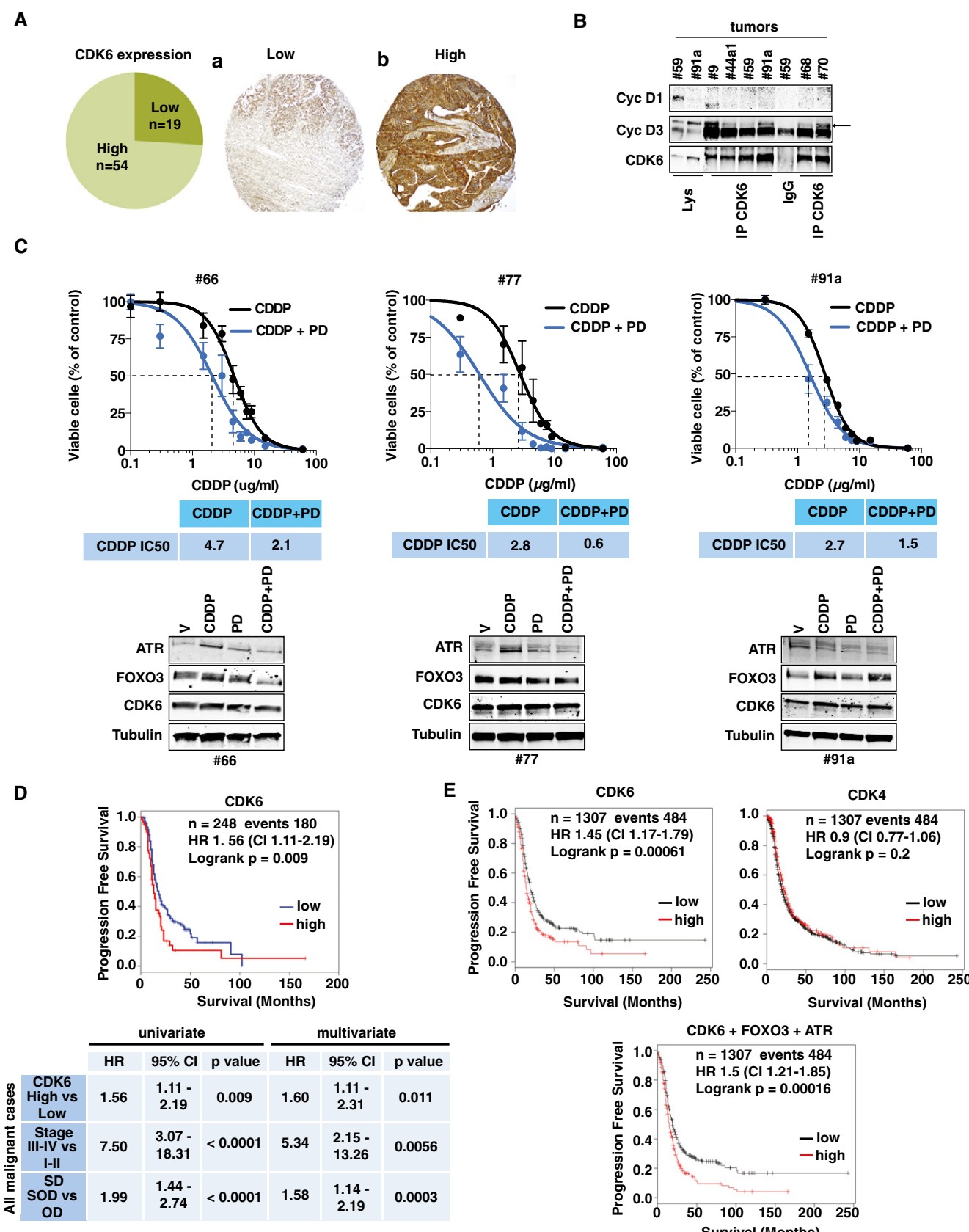

Figure 7.

# Materials and Methods

### Study approval

Our institutional Biobank has collected samples and obtained informed consent from all patients. The Internal Review Board approved this study (#IRB-06/2011). Animal experiments were approved by the Italian Ministry of Health (#1261/2015-PR) and by Institutional Animal Care and Use Committee (OPBA) and conducted according to that committee's guidelines.

### Primary EOC collection and analyses

Human EOC samples were immediately frozen and stored in liquid nitrogen until needed. Fresh tumor specimens were used to generate primary EOC culture. Frozen EOC samples were lysed in RIPA buffer for Western blot and immunoprecipitation analyses as described (Sonego *et al*, 2013). Tissue microarray (TMA) was built taking the most representative areas from each tumor using a semiautomatic tissue arrayer instrument (Galileo CK3500 TMA, ISENET), as described (Perrone *et al*, 2016). TMA sections were stained with primary anti-CDK6 (Santa Cruz) followed by biotinylated IgG (Dako LSAB2 System).

### Vectors, transfections, and recombinant viruses

pCMV-CDK6 and pEGFP-histone-H1 vectors were described (Baldassarre *et al*, 1999, 2000; Sonego *et al*, 2013). pFLAG-FOXO3 was a gift from K. Guan (#13507 Addgene Inc.) and was used to generate the GFP-FOXO3 vectors, using the pEGFP-C1 vector (Clontech). CDK6 and FOXO3 point mutants were generated using the QuikChange Site-Directed Mutagenesis Kit (Agilent). GST-FOXO3 deletion mutants were generated by cloning FOXO3 PCR fragments in the pGEX vector. pLKO shRNAs (Sigma) and primers are listed in Appendix Table S3.

### Reagents

Carboplatin (CBDCA) and cisplatin (CDDP) (TEVA Italia) were used for *in vitro* and *in vivo* experiments. PD0332991-HCl (PD) was purchased from SelleckChem (S1116), cycloheximide (CHX, C4859) from Sigma, and MG132 (474791) from Calbiochem.

### Cell culture and generation of platinum-resistant cell line

MDAH-2774, OV90, SKOV-3, OVCAR3, ES2, CAOV3, TOV112D, TOV21G were purchased from ATCC; OVSAHO and KURAMOCHI from JCRB (Japanese Collection of Research Bioresources); OVCAR8 and OVCAR4 from NCI; and COV362 from ECACC; IHEOC were purchased from ABM (Applied Biological Materials). Mycoplasma contamination was assessed every 15 days using the MycoAlert test (Lonza). Cell lines were authenticated according to the Cell ID TM System (Promega) protocol and using Genemapper ID Ver 3.2.1. Cells were kept frozen, and used no later than 2 months in culture for performing the experiments.

Epithelial ovarian cancer cell lines were maintained in RPMI-1640 medium (Sigma-Aldrich Co.); immortalized human ovarian epithelial cells (IHOEC) in Pigrow I medium (Applied Biological Materials); 293FT and 293T17 in DMEM medium (Sigma-Aldrich Co.), all supplemented with 10% heat-inactivated FBS

(Sigma-Aldrich Co.). Cell lines were authenticated according to the Cell ID TM System (Promega) protocol and using Genemapper ID Ver 3.2.1. SKOV3ip cells are SKOV3 cells adapted to *in vivo* growth by recovering them from intra-peritoneal injection, as described (Yu *et al*, 1993). Platinum-resistant cell lines were generated by treating MDAH cells for 2 h with a CDDP dose 10-fold higher than the IC50, followed by a recovery period. After 20 cycles of CDDP treatment, the resulting cell population was maintained in drug-free medium. The new IC50 was calculated comparing them to MDAH parental cells (Sonego *et al*, 2017).

### Loss-of-function screening

Lentiviruses expressing shRNAs were produced by transfecting 293FT cells with pLP1, pLP2, pVSV-G (Invitrogen), and pLKO vectors (Sigma) in 96-well plates and used to transduce MDAH cells plated in 96-well plates. Seventy-two hours later, cells were treated with 140 μg/ml CBDCA for 16 h. Cell viability was evaluated 24 h later using CellTiter 96 AQueous kit (Promega).

### Cell viability, IC50 drug calculation, and colony assay

CBDCA and CDDP IC50 in parental and shRNA-transduced cells are reported in each figure and were calculated using increasing doses of platinum for 16 h. PD IC50 was calculated by exposing cells to increasing concentration of PD (ranging from 0 to 100 μM) for 24 h. Calculated IC50 were as follows: 40 μM for OVSAHO and KURAMOCHI; 30 μM for SKOV3ip and 20 μM MDAH and OVCAR8. Combination treatments using platinum ± PD were performed by treating cells with increasing doses of platinum for 16 h and then with PD for 24 h using half of the calculated IC50, in each cell line. Combination index was calculated using CDDP and PD at the calculated IC25, IC50, and IC75, alone and in combination. Data were analyzed using the Calcusyn software. In all experiments, cell viability was determined 24 h after the end of treatment, using the CellTiter 96 AQueous cell proliferation assay kit (Promega).

### Colony assay

MDAH and SKOV3ip cells were silenced using control- or CDK6-shRNAs on day 1, then plated on day 2 in 6-well plate, and treated on day 3 with CDDP (0.3 μg/ml for 72 h). Cells were allowed to recover and grow for 3 more days, before staining with crystal violet. For colony assay comparing MDAH parental and platinum-resistant clones, cells were seeded in 6-well plates mixing parental and platinum-resistant MDAH cells at different ratios, as indicated. Cells treated with CDDP ± PD0332991 were allowed to grow for 10 days before staining with crystal violet. After taking pictures, the absorbance of crystal violet dissolved in 1% SDS PBS solution was read at 595 nm to quantify cell growth.

### Growth curve and FACS analyses

For growth curve analyses, SKOV3ip and OVCAR8 cells were plated in 6-well plates ($50 \times 10^3$ cells/well), transduced with the indicated shRNAs, and, starting 24 h post-transduction, counted

daily by trypan blue exclusion test. MDAH cells stably transduced with control- or CDK6-shRNA were plated on 6-well ($50 \times 10^3$ cells/well) and counted daily by trypan blue exclusion test.

For FACS analyses, MDAH, SKOV3ip, and OVCAR8 cells were plated in 100-mm dish and silenced with specific shRNAs or treated as indicated. Cells were then collected with trypsin, fixed in ice-cold 70% ethanol, washed twice in PBS 1×, and resuspended in propidium iodide (50 μg/ml supplemented with 100 μg/ml RNase A in PBS). Then, cells were subjected to FACS analysis using a FACScan instrument (BD Biosciences). Distribution of cells in subG1, G1, S, and G2/M phases was calculated using the WinMDI2.8 software. PD was used at the concentration of 8 μM for MDAH and OVCAR8 and 15 μM for SKOV3ip cells. CDDP was used at the concentration of 0.75 μg/ml in OVCAR8, 1.5 μg/ml in MDAH, and 3 μg/ml in SKOV3ip, for 16 h.

### SA-β-galactosidase assay

Senescence assay was performed as described (Anders *et al*, 2011). Cells were fixed with 2% formaldehyde and 0.2% glutaraldehyde solution for 20 min at 4°C and then incubated overnight at 37°C in X-gal solution. Blue positive cells were counted in 10 randomly selected fields using a 40× objective.

### Immunofluorescence and video time-lapse microscopy

For immunofluorescence analyses, cells plated on coverslips and fixed in PBS 4% paraformaldehyde (PFA) or tissue sections from paraffin or frozen samples, as indicated, were stained with primary antibodies, as reported (Berton *et al*, 2014). γH2AX$^{S139}$, pRB1$^{S780}$, Ki67 and FOXO3 (described below), propidium iodide (Sigma) or TO-PRO-3 (Invitrogen) were used to stain the nuclei. Stained cells were analyzed using a confocal laser-scanning microscope (TSP2 or TSP8, Leica) interfaced with a Leica fluorescent microscope. Collected images were analyzed using the LAS (Leica) and the Volocity® (PerkinElmer) software.

For time-lapse microscopy, MDAH cells were seeded in 12-well plate ($20 \times 10^3$ cells/well) or on glass bottom dishes (20,000 cells) (WillCo Wells BV) transfected with GFP-histone H1 and treated with CBDCA (140 μg/ml) and/or PD0332991 (8 μM). Images were collected every 5 min for 10 h using a 20× or a 63× immersion glycerin objective using a time-lapse AF6000LX workstation (Leica) at 37°C with controlled humidity and $CO_2$ concentration. Video was generated assembling the images with the Volocity® software (PerkinElmer), as described (Belletti *et al*, 2010; Sonego *et al*, 2013; Fabris *et al*, 2015).

### Protein stability

FOXO3 protein stability was evaluated by treating (or not) with CDDP (15 μg/ml for 3 h) shRNAs-transduced or shRNAs-transfected (with GFP-FOXO3$^{WT}$ or GFP-FOXO3$^{3S325A}$) MDAH cells and then released in medium with CHX (10 μM) for 2, 4, 6, or 8 h, as described (Lovisa *et al*, 2016). Inhibition of FOXO3 proteasomal degradation was evaluated by treating 293T17 cells with CDDP and with or without MG132 (10 μg/ml) for 3 h. FOXO3 expression was evaluated by Western blot.

### Mitotic spreads and PCC evaluation

MDAH cells, transduced and treated as indicated, were incubated 10 min at 37°C in 75 mM KCl solution, resuspended in Ibraimov's solution, and then fixed in methyl alcohol:acetic acid (3:1) solution. Fixed cells were spread on a glass slide, dried, and stained with DAPI (Sigma). A fluorescence microscope was used to count normal mitosis and PCC. At least 100 nuclei were analyzed in each condition.

### *In vivo* analyses

All animal experiments were performed following validated procedures (Sonego *et al*, 2013; Vecchione *et al*, 2013) approved by the Italian Ministry of Health (#1261/2015-PR). Female athymic nude mice of 5 weeks (Hsd: Athymic nude Foxn1 nu/nu, Harlan Laboratories, median weight 21 g) were maintained in our conventional animal facility at 22°C with 40–60% of humidity according to the Animal Care and Use Committee of the Centro di Riferimento Oncologico indications and following the 3R guidelines. SKOV3ip and MDAH xenografts were established by injecting subcutaneously $2 \times 10^6$ cells in both flanks of mice. When tumors reached 40–60 mm$^3$, a cohort of three to seven mice for each group (vehicle, CBDCA, PD, or CBDCA + PD) was treated with suboptimal concentration of CBDCA (20 mg/kg) and PD0332991 (150 mg/kg), alone or in combination. Mice were observed and tumors measured every 2 days for 16–18 days. Experimental end point was set according to Institutional and Italian Ministry of Health guidelines. Xenografts from pre-transduced MDAH were established by injecting cells 48 h post-transduction in right (sh-ctrl) and left (sh-CDK6) flanks of nude mice ($2 \times 10^6$ cells/flank). A cohort of six mice for each group (vehicle or CBDCA) was treated with CBDCA as described above. Animals were sacrificed by $CO_2$ inhalation euthanasia, and explanted tumors were frozen and included in paraffin for subsequent analyses. Sample size was calculated using the G*Power tool based assuming that type I error is equal to 5% and power equal to 80% ($\alpha = 0.05$ and $\beta = 0.20$).

### Immunoblotting

Extraction of total proteins, Western blot, and immunoprecipitations analyses were performed as described (Sonego *et al*, 2013; Fabris *et al*, 2015; Lovisa *et al*, 2016). Primary antibodies used were as follows: vinculin (N-19, 1:1,000), CDK6 (C-21, sc-177, 1:800), CDK4 (C-22, sc-260, 1:400), pRB1S780 (sc-12901, 1:400), PSF (39-1, sc-101137, 1:800), CHK1 (G-4, sc-8408, 1:500), luciferase (sc-32896), lamin A (C-20 sc-6214) (Santa Cruz Biotechnology); FOXO3 (75D8, #2497, 1:600), cyclin D3 (DCS22, #2936, 1:500), pCHK1S296 (#2349, 1:500), H2AX (D17A3, #7631, 1:500), ATR (#2790, 1:500), ATM (#2873, 1:500), and cleaved caspase-3 Asp175 (#9661, 1:500) (Cell Signaling); cyclin D1 (DCS-6, CC12, 1:1,000) (Millipore); γH2AXS139 (#2577, 1:500) (Upstate Biotechnology); RB1 (554136, 1:500) and GRB2 (610111, 1:300) (BD Biosciences); actin (A5060, 1:500), tubulin (T5168, 1:1,000), Flag (F3040), OP18 (O0138, 1:1,000), and V5 (A7345)- and HA (A2095)-agarose conjugated (Sigma-Aldrich Co); GFP (11 814 440 001, 1:500) (Roche); Ki67 (ab15580, 1:1,000) (Abcam). Quantification of the blots was done

using the QuantiONE software (Bio-Rad Laboratories) or the Odyssey infrared imaging system (LI-COR Biosciences).

### Recombinant protein and *in vitro* kinase assay

Production of FOXO3 recombinant proteins was performed by cloning cDNA fragments in pGEX vector and by transforming *E. coli* BL21 pLys bacteria with the expression plasmid. Recombinant proteins were purified with glutathione Sepharose resin (GE Healthcare) as previously described (Berton *et al*, 2014; Fabris *et al*, 2015). *In vitro* kinase assays were performed by incubating cyclin D3-CDK6 or cyclin D1-CDK4 active kinase complexes (Signal Chem) with GST-tagged FOXO3a full length (Abnova), GST-FOXO3 mutants, or GST-RB1 recombinant proteins, as substrates, for 30 min at 30°C in the presence of 1.5 μCi of $\gamma^{32}$P-ATP (PerkinElmer Life Sciences), as described (Berton *et al*, 2014; Fabris *et al*, 2015).

### qRT–PCR analyses

qRT–PCR analyses were performed essentially as described (Sonego *et al*, 2013; Berton *et al*, 2014; Fabris *et al*, 2015). Briefly, total RNA was extracted using SV 96 Total RNA Isolation System kit (Promega) or Trizol reagent (Invitrogen), quantified using NanoDrop (Thermo Fisher Scientific Inc., USA), and retro-transcribed using the Go-Script reverse transcriptase (Promega) or the AMV reverse transcriptase (Promega). cDNAs were amplified using SYBR green dye-containing reaction buffer (Experteam) and the MyiQ2 Two Color Real-time PCR Detection System (Bio-Rad). Normalization of the data was performed using the POL2A expression.

### Chromatin immunoprecipitation

Chromatin immunoprecipitation protocol was adapted from Carey *et al* (2009). Briefly, cells treated with 1% of formaldehyde were lysed and chromatin sheared by sonication. After IPs, DNA was purified and analyzed by qRT–PCR. Results were reported as ATR enrichment, meaning the ratio between SQ (Starting Quantity) of IP of each ATR promoter fragment and SQ of unrelated IP, as described (Schoppee Bortz & Wamhoff, 2011).

### Statistical analysis

Statistical significance (*P* < 0.05), means, standard deviation, 95% confidence intervals were determined by using GraphPad PRISM software (version 6.0f) using the most appropriate test, as specified in each figure. Exact *P*-value of each experiment is reported in Appendix Table S4.

For public EOC microarray data analysis, raw data were downloaded from GEO (GSE9891) (Kollmann *et al*, 2013) and preprocessed using robust multiarray analysis (RMA) method (McCall *et al*, 2010), as implemented in the firma package of Bioconductor (Gentleman *et al*, 2004). Quality control was applied on raw and normalized data. All statistical tests were two-sided. Analyses were conducted in R environment version 3.0.2.

Combination index (CI) to measure whether two drugs have a synergistic (CI < 1), additive (CI = 1), or antagonistic effects (CI > 1) was performed applying the Chou–Talalay method, using the Calcusyn software.

## The paper explained

### Problem

Epithelial ovarian cancer (EOC) of high grade represents the most lethal gynecological malignancy. The prognosis of EOC patients mostly relies on the response to platinum-based therapy. Although more than 70% of EOC patients respond to the first line therapy, most of these patients relapse and develop platinum-resistant, largely incurable, disease. Therefore, identifying new therapeutic options able to improve platinum efficacy represents a clinical and scientific challenge that could greatly impact on patients' progression-free and overall survival.

### Results

Using a functional genomic screening, we discovered that genetic silencing of CDK6 increased EOC cells sensitivity to platinum and demonstrated that CDK6 kinase activity is necessary to protect EOC cells from platinum-induced death. An orally available inhibitor of CDK4/CDK6 kinase activity (PD0332991) sensitized EOC cell line and primary culture to platinum and significantly inhibited EOC *in vivo* growth when administered after platinum as maintenance therapy. Functional and biochemical analyses showed that, upon platinum treatment, cyclin D3-CDK6 complex binds and phosphorylates the transcription factor FOXO3 on serine 325, increasing its stability and regulating the expression of ATR. Blockage of CDK6/FOXO3 axis results in EOC cell death, due to ATR/CHK1 axis inhibition that drives cells blocked in S phase by platinum to premature chromosome condensation. In line with these observations, high CDK6 expression levels predicted poor survival of EOC patients.

### Impact

Several CDK4/6 inhibitors (such as PD0332991) are currently tested in cancer patients as single agent or in combination with chemotherapy or targeted therapy, with promising results. Our data, showing that CDK6 represents a critical target in EOC response to platinum and that it is overexpressed in patients with worse prognosis, are therefore of primary translational relevance. We suggest that selected EOC patients could benefit from a combined therapy based on platinum and CDK4/6 inhibitors. In particular, our data suggest that adding CDK4/6 inhibitors as maintenance therapy after platinum could prove particularly effective in recurrent patients who did not completely respond to the first line platinum-based therapy. Overall, this work unveils a therapeutic strategy to increase platinum efficacy in EOC patients that could be exploited to improve the clinical response and the survival of these patients.

For survival analysis, patients were grouped based on similar clinical and pathological characteristics. Stage and surgical debulking were coded as dichotomous indicator variables, that is, stage III/IV vs. stage I/II, suboptimal debulking (SOD) vs. optimal debulking (OD). For CDK6, genes, patients were classified in high vs. low expressing patients, according to the relative higher quartile. PFS was identified as the relevant clinical end point. PFS curves were reported according to the Kaplan–Meier method and were compared with the log-rank test. Median estimates, with 95% confidence intervals (CI), were also reported.

For each relevant prognostic variable, a Cox univariate model was used to estimate the hazard ratio (HR). Multivariable analysis was performed using a Cox regression model to evaluate the prognostic impact in the context of concomitant effects of other known clinical prognostic factors (stage and residual disease).

Kaplan–Meier survival curves were generated using the KM Plotter online tool (http://kmplot.com), using the most appropriate cutoff choice. KM Plotter is an online algorithm exploitable to interrogate the expression of up to 54,675 genes on up to 1,648 ovarian cancer patients, with a mean follow-up of 40 months.

**Expanded View** for this article is available online.

## Acknowledgements
We thank Dr. M.S. Nicoloso and Dr. R. Spizzo for critical reading of the manuscript and all members of the SCICC Lab for supportive scientific discussions. We thank Dr. K. Guan for FOXO3 plasmids. This work was supported by grants from Ministero della Salute (RF-2010-2309704), Associazione Italiana Ricerca sul Cancro (AIRC) (IG 12854) and CRO 5X1000 funds to GB and by CRO-5X1000 YIP grants to MS.

## Author contributions
GB had the original idea. GB, ADA, and MS designed the experiments. GB, ADA, MS, and BB analyzed the data and wrote the manuscript. ADA performed most of the experiments. MS, IPelli, IPella, SDA, and SB participated in some experiments. MS and IPelli generated the MDAH platinum-resistant cell clones. VC performed all the pathological analyses. RS, GG, and LM provided patients' data and samples for expression analyses and primary cell cultures. MB and DM provided survival analyses using available datasets. GB and ADA performed the analyses using the KMP Web sites. DC and GC constructed, colored, and analyzed the TMA data. JA identified the putative FOXO3 binding sites on the ATR promoter, using bioinformatic tools. All authors read and approved the manuscript.

## Conflict of interest
The authors declare that they have no conflict of interest.

## For more information
Addgene: http://www.addgene.org/
Addgene is a non-profit plasmid repository dedicated to helping scientists around the world share high-quality plasmids.
GPS3.0 (Group-based Prediction System, version 3.0): http://gps.biocuckoo.org/
GPS3.0 is a prediction software available online that could predict kinase-specific phosphorylation sites for 408 human PKs in hierarchy.
KMP (Kaplan–Meier Plotter): http://kmplot.com/
KMP is a web application for assessing the effect of 54,675 genes on survival using 10,188 cancer samples.

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
