## [Review Process File · EMBO Molecular Medicine]

CDK6 protects Epithelial Ovarian Cancer from platinum-induced death via FOXO3 regulation

Alessandra Dall'Acqua, Maura Sonego, Ilenia Pellizzari, Ilenia Pellarin, Vincenzo Canzonieri, Sara D'Andrea, Sara Benevol, Roberto Sorio, Giorgio Giorda, Daniela Califano, Marina Bagnoli, Loredana Militello, Delia Mezzanzanica, Gennaro Chiappetta, Joshua Armenia, Barbara Belletti, Monica Schiappacassi and Gustavo Baldassarre

*Corresponding authors: Monica Schiappacassi and Gustavo Baldassarre
CRO Aviano, National Cancer Institute*

Review timeline:	Submission date:	30 August 2016
	Editorial Decision:	20 October 2016
	Author's Appeal:	27 October 2016
	Editorial Decision:	18 January 2017
	Revision received:	20 June 2017
	Editorial Decision:	07 July 2017
	Revision received:	10 July 2017
	Accepted:	11 July 2017

Transaction Report:

Editor: Roberto Buccione

1st Editorial Decision

20 October 2016

I apologise for the delay in providing you with a decision. In this case we experienced unusual difficulties in securing three willing and appropriate reviewers. Unfortunately, I have not been able to obtain a third evaluation on this manuscript in a timely fashion. Hence, I have decided to proceed based on the two evaluations to avoid further delays.

As you will see, while Reviewer 2 is more supportive albeit not enthusiastic, Reviewer 1 is much more reserved with fundamental and serious concerns. In aggregate, I am afraid the issues raised preclude publication of the manuscript in EMBO Molecular Medicine. I will not discuss each point in detail as they are clearly stated.

The many concerns, some of which indeed overlapping between the reviewers, include limited novelty, insufficient in vivo modelling, unclear case for (novel) translational value and insufficient mechanistic development.

Further reviewer cross-commenting highlighted the fact that the used cell line models are not representative of HGS-EOC that could theoretically benefit from the combination of platinum drugs and CDK6-targeted therapy and the limited efficacy observed, which limit the translation and clinical implications of your work. Finally, the fundamental lack of significant novelty was

reiterated.

Given these fundamental concerns and the overall lack of enthusiasm, I have no choice but to return the manuscript to you at this stage. In our assessment, it is not realistic to expect to be able to address these issues experimentally and to the satisfaction of the Reviewers in a reasonable time frame.

I am sorry to have to disappoint you at this stage, however. I hope that the Reviewers' comments will be helpful in your continued work in this area.

***** Reviewer's comments *****

Referee #1 (Comments on Novelty/Model System):

Technical Quality As detailed in the remarks to Authors, some experiments are poor or inadequately reported

Novelty: Almost all information is not novel, as detailed in the remarks to Authors.

Medical impact: is low because CDK6 proposed as target is highly expressed and fruitfully inhibited only in the cell line studied. Moreover, dual CDK4/CDK6 inhibitors have undergone trials and are already undergoing further trials.

Adequacy of model system: almost all results have been obtained in one cell line, that is not an adequate model of high-grade serous epithelial ovarian carcinomas. Moreover, the only alternative cell line used is more inadequate.

Referee #1 (Remarks):

The paper submitted by Dell'Acqua et al. reports the results of a screening with 23 shRNAs targeting CDKs in one ovarian cancer cell line, that leads to the identification of CDK6 as the most active in hampering the effectiveness of carboplatin. As carboplatin is the drug used commonly as first line treatment of human epithelial ovarian carcinoma and refractoriness/resistance to platinum drugs is the major determinant of patient relapse and eventually death, they conclude that targeting CDK6 might improve patients' response. They complete the work with several experiments to investigate the mechanism of CDK6 involvement in cell survival and report a non-canonical effect of CDK6. The information is interesting and well documented, but overall it lacks broad significance and novelty.

Almost all experiments have been carried out only in one cell line (referred to as MDAH and presumably being MDAH-2774, although the correct full name is never mentioned in any section of the MS). Only a few experiments have been repeated using OVCAR8 cells. MDAH-2774 cells are derived from an endometrioid ovarian carcinoma, showing MSI, while OVCAR8 cell have been classified as derived from a clear cell ovarian carcinoma. Therefore both cell lines are poorly representative of the high-grade epithelial ovarian carcinomas which represent the majority of ovarian cancers, of whom platinum sensitivity and refractoriness/resistance have been thoroughly studied. Additionally, while OVCAR8 cells have been thoroughly characterized, and found a typical CC EOC as mentioned above, the information on MDAH (2774?) cells is poorer and only the mutational status of TP53 was reported. It noteworthy that the existence of high-grade endometrioid ovarian carcinomas is questioned, while the low grade endometrioid, CCC and other type I EOC, are cured in 90% of cases with surgery alone and do not respond to platinum drugs.

As the Authors themselves show, MDAH cells express the highest level of CDK6 among several ovarian cancer cell lines examined. This presumably demonstrates these cells' unique dependence on this kinase. Only in another cell line (OVCAR8) the addition of exogenous constitutively active CDK6 results in increased survival of cells treated with platinum drugs. The latter cell line, where CDK6 and Rb are poorly expressed, as well as other two cell lines (Kuramochi and OVS9), similarly showing low levels of CDK6, are insensitive to Palbociclib alone. This information as well as the other experiments strongly suggests a stringent context specific effect of the CDK6 inhibitor, also on platinum sensitization, restricting the therapeutic opportunities. This is in line with other reports showing cell type specific efficacy of CDK inhibitors, for example of dinaciclib in

sensitizing to platinum only ovarian cancer cells showing certain genotypes (see e.g. Chen et al., *Oncotarget* 2015, not quoted).

Trials are ongoing already with dual CDK4/CDK6 inhibitors, including those with Palbociclib, in ovarian cancer. Moreover another dual inhibitor that came out to be more effective in patients (including those with ovarian cancer) (Patnaik et al. *Cancer Discovery* 2016) and less toxic, has undergone phase I trial and is ongoing further trials. To my knowledge CDK6 specific inhibitors do not exist and anyway, according to the data reported in this manuscript, they would be effective only in tumors showing CDK6 high level of expression, if any. Therefore, clinical data will be available soon, which will overcome the interest of preclinical data in one cell line.

The non canonical activities of CDKs, that phosphorylate transcription factors such as FOXO1 (Huang et al *Science* 2006, not quoted) and FOXM1 (quoted) have been reported already and recently reviewed (Hydbringe et al. *Nat Rev Mol Cell Biol* 2016). Thus, to my knowledge the only novelty of this work is the demonstration that FOXO1 regulates the expression of ATR and, through ATR, DNA damage response. This information might be interesting for a more specialized journal.

Minor comments

The colony assay of Figure 2C and D is poorly convincing, not only for the results (this reviewer does not appreciate any difference in panel 2C to explain results of panel 2D), but also because the MSI of MDAH cells makes the results difficult to interpret.

In vivo experiments are short-term: the real tumor volumes should be measured after longer inspection and shown on the y axis, rather than reported as fold increase.

The possible platinum dependent increase of CDK6 and FOXO3 interaction is mentioned and documented but not explained and discussed.

Referee #2 (Remarks):

Platinum-resistance is a major issue in the treatment of HGEOC. This manuscript highlights several interesting findings seeking to elucidate novel drugs and therapeutic targets for the treatment of platinum resistant HGEOC. The authors show that CDK6 inhibition is synthetic lethal with platinum and platinum+ CDK6 inhibitor combination was effective in preventing the re-growth of platinum-resistant cells. Additionally, the authors propose that the effects of CDK6 inhibition are mediated through the regulation of FOXO3 and ATR. Though there are some interesting and novel findings in this manuscript there are several experiments that are required in order to strengthen the claims proposed by the authors.

- 1) For the initial shRNA screen, was there confirmation of knockdown for all 23 CDK's?
- 2) The authors mainly use the MDAH and OVCAR8 cell lines but the results are not consistently presented, ie, in Fig 1E MDAH shows CDK6 expression for just one day but OVCAR8 show the expression for 3 days, also in Fig1F the senescence assay was only performed in MDAH and not OVCAR8. It would be interesting to see whether the same effects are seen in OVCAR8 which actually seem to be more responsive to shCDK6 (Fig 1D).
- 3) Colony formation assays should be performed with the platinum+shCDK6 in order to assess the effects of long-term exposure to platinum as well as assessing apoptosis.
- 4) In order to confirm that CDK6 is a potent target in the platinum resistant setting, overexpression experiments should be performed in order to assess whether increased expression of CDK6 decreases sensitivity to platinum. Furthermore, if CDK6 is driving resistance its expression should be increased in platinum-resistant cells therefore the expression of CDK6 should be determined in the platinum-resistant clones generated from the MDAH cells.
- 5) In order to confirm that platinum + the CKD6 inhibitor (PD) is an effective combination therapy the authors should calculate the combination index or perform an isobologram analysis.

6) Though the authors mention that the effects of the PD compound are CDK6-dependent this can not be determined unless re-expression of CDK6 in the low expressing cells sensitize the cells to PD or direct inhibition of CDK6 using CRISPER or shRNA leads to PD resistance.

7) The in vivo studies using platinum + PD are promising but not dramatic and Fig 3B should be plotted as tumor volume not fold change over time in order to confirm that all tumors started treatment at a similar size. Furthermore, it is not clear whether Fig 3C is tumor volume at the end of the study, if so, the tumors were still very small when the study ended therefore it would be important to show what size the tumors were when they started treatment.

8) It is clear that FOXO3 and CDK6 bind but the half-life studies are not significant: in Fig. 5A it would be expected that the sh-CDK6 should start off with less FOXO3 at time point 0. Can the authors explain why we do not see a decrease at this time point?

9) Lastly, for the combination studies in the primary cell lines, there are clearly differences in response that cannot be attributed to the endogenous levels of CDK6, FOXO3 or ATR. Have the authors assessed the levels of these proteins post-treatment? The authors need to determine whether these cell lines behave the same way was MDAH and PD + platinum treatment results in degradation of CDK6, FOXO3 and/or ATR.

Authors' Appeal

27 October 2016

Thank you for your e-mail regarding our manuscript EMM-2016-07012.

We are of course very disappointed with the final outcome of the reviewing process that was also particularly long.

I am well aware that your editorial decision is based not only on the comments we received but also on several other considerations you reached, possibly based on reviewers' confidential comments and priority ranking and so on.

Although I think that it will be probably useless writing this rebuttal, I also think that it is necessary to highlight our concerns on Reviewer 1 comments. These indeed appear largely based on personal considerations, often in contradiction with current literature on ovarian cancer and on pathways studied in the manuscript.

I wish to bring to your attention several specific points that, in my opinion, merit your careful consideration and that may also convince you to re-consider the appropriateness of Reviewer 1' comments.

1. The first concern expressed by Reviewer 1 is the overall novelty of the manuscript.

However, along the revision, He/She makes continuous confusion between FOXO1, FOXO3 and FOXM1 that, although members of the same gene family, are different genes that produce different proteins and that have different functions. So, the fact that it has been previously reported an interaction between CDK4 and FOXO1 or FOXM1 does not diminish the novelty of our original observation that CDK6 (and not CDK4) binds and phosphorylates FOXO3.

Other novel points of our manuscript are:

1a. The demonstration that CDK6 regulates the DNA damage response in a cell cycle independent way

1b. The demonstration of the biological significance of CDK6/FOXO3 interaction in ovarian cancer cells response to platinum

1c. The detailed molecular description of the mechanism by which CDK6 regulates FOXO3 expression and protein stability

1d. The identification of the serine in FOXO3 specifically phosphorylated by CDK6 in complex with Cyclin D3 (and you probably know that this is not a trivial achievement!)

1e. The identification of FOXO3 binding sites on ATR promoter and the new description that FOXO3 regulates ATR (this point acknowledged also by Reviewer 1)

- 1f. The first description that CDK6 is overexpressed in about 70% of primary ovarian cancer
- 1g. The first description that Cyclin D3 expression increases in platinum-treated ovarian cancer samples
- 1h. The first formal demonstration that CDK6 does not share with CDK4 the ability to phosphorylate a member of the FOXO gene family
- 1i. The demonstration that a sequential administration of CDK4/6 inhibitors after platinum is an effective and feasible therapeutic approach in ovarian cancer. This is a new and important point, toward the future development of new anticancer therapies based on the combined use of chemo- and targeted-therapies
- 1j. The first in silico demonstration that combined expression of CDK6 and FOXO3 mRNA expression predicts prognosis of ovarian cancer patients.

2. Regarding the models used we are in complete disagreement with Reviewer 1, for several reasons.

First, it is largely accepted that High-grade Endometrioid cancer exists and shares with HG-Serous and undifferentiated OC similar clinical behaviour and molecular alterations (Lancet, 384, No. 9951, p1376–1388, 2014. Am J Pathol. 186:733-47, 2016). Actually, it is not rare that high grade with serous and endometrioid areas coexist in the same ovarian cancer. These are very well known pathological entities known as mixed epithelial ovarian cancer. So it is more than appropriate to refer to this group of neoplasias as HGEOC (High Grade Epithelial Cancer), as we did.

The models we used to study HGEOC response to platinum are thus very appropriate to study this complex pathology.

MDAH 2774 (MDAH) cells carry mutations in both p53 (R273H) and BRCA2 (D1784fs Ter3) genes that represent the most typical mutations of HGEOC.

OVCAR8 cells are classified as highly probably HGSOC cells by a recent report, based on their mutational spectrum, copy number alteration and gene expression profile (Nat. Comm. 4, Article number: 2126 (2013) doi:10.1038/ncomms3126)

We used OVCAR8 for a very precise reason: they are RB null cells and RB deletion is a typical genetic alteration of recurrent HGSOC, as recently reported by Patch et al. (Nature. 521:489-94, 2015). The fact that OVCAR8 cells are RB null was instrumental to demonstrate that we were looking to an RB-independent effect of CDK6.

Furthermore, we confirmed our observations on primary cultures from HGSOC patients and this kind of approach represents, to our knowledge, the state of the art of preclinical research in cancer.

3. It is simply untrue and misleading the concept that low grade ovarian cancer are cured by surgery alone.

Surgery alone is only partially curative, for all ovarian cancer types of stage I, independently from their grade (Br J Cancer. 105:1818-24, 2011).

For later stages all ovarian cancer histotypes are treated with platinum-based therapies and, among all histotypes, clear cell carcinomas are the most platinum-resistant (Gynecologic Oncology 109, 370–376, 2008). All past clinical trials testing new drug combinations and most of the ones that are currently actively enrolling ovarian cancer patients include the possibility to have tumors of different histotypes.

Therefore, all we know, in terms of response to chemotherapy of ovarian cancer, is based on studies in patients with similar stages (e.g. stage I-II vs III-IV) but with all histotypes, likely represented for their frequency among epithelial ovarian cancer (roughly 70% serous, 15% Endometrioid, 10% Clear Cell and 5% Undifferentiated).

Thus, even if Reviewer 1 was right in defining OVCAR8 cells as a model of Clear cell carcinoma, the clinical significance of our work was not diminished but rather increased by the use of this model.

I would also like to highlight that most of the recent and past literature on Ovarian Cancer and HGSOE is based on models that do not match the criteria to be defined as HGSOE (Nat. Comm. 4, Article number: 2126 (2013) doi:10.1038/ncomms3126).

One of the most commonly used cell line is SKOV3. We have in the lab these cells and we tested them for their sensitivity to CDK6 inhibition. SKOV3 behaved as the other cell lines but we excluded them from this manuscript exactly to avoid the possibility that a Reviewer could raise some concerns on their use as model of HGSOE.

A recent relevant publication by the group of Drapkin highlights the importance in ovarian cancer research, and in particular in HGSOE, to balance the necessity of using a representative model with the necessity of using a model that allows the experiments to be carried out (Gynecol Oncol. 139:97-103, 2015).

This is exactly what we have tried to do. We think that careful reading of our work getting rid of preconceptual positions on “cellular models”, that at the end remain only models and not more not less than that, would lead to a different opinion on our results.

4. Finally, regarding the translational relevance of our work and the fact that CDK4/6 inhibitors are in clinical development. This is news that we perfectly know and that, as discussed in the manuscript, in our opinion strengthen the possible clinical application of our preclinical results. Specifically:

CDK4/6 inhibitors are generally used as single agents and their association with chemotherapy has never been tested in humans. Based on our data, it would be worth testing this possibility in HGSOE patients.

To our knowledge, the only clinical trial ongoing to date in ovarian cancer patients is the testing of Palbociclib as single agent in patients overexpressing Cyclin D1 and expressing RB. These patients, based on our data do not represent the ones most likely sensitive to the combination of CDK6 inhibition and platinum. However, this trial is not testing the efficacy of Palbociclib in association with chemotherapy in ovarian cancer.

It is true that CDK6 specific inhibitors do not exist. This evidence does not affect the significance of our work since we demonstrated that the dual CDK4/6 inhibitor could be used with platinum to enhance its activity. However, the different molecules now in clinical development as CDK4/6 inhibitors including Palbociclib, LE001 and Abemaciclib inhibit with different relative IC50 CDK4 and CDK6, supporting the possibility that more specific inhibitors for each kinase could be developed in the future if it will be necessary to increase their specificity and/or to decrease toxic side effects.

Overall we believe that a revision all based on personal conviction and considerations and not on the careful evaluation of the presented data and literature is not the one we could expect from a highly qualified journal as EMBO Molecular Medicine.

2nd Editorial Decision

18 January 2017

Thank you for the submission of your manuscript to EMBO Molecular Medicine. We have now heard back from the reviewers whom we asked to evaluate your manuscript. I am truly sorry for the significant delay in providing you with a decision. Although I cannot do much to mitigate your understandable frustration, I can at least try to explain why.

First of all, there is the history of your manuscript and my commitment to obtain good quality evaluations with a view to overruling the admittedly unbalanced evaluation by reviewer 1, as also discussed with an expert external advisor. I consequently tried to recruit additional very expert scientists with specific expertise to evaluate your manuscript. They were not immediately available and therefore I decided to give them extra time rather than settle for less experienced ones. Furthermore, the overlap with the holiday season did not help. Finally, we are currently still dealing with the aftermath of the holiday season in terms of backlog and your case required extra time to re-

analyse it and re-discuss with my colleagues.

You will see that I eventually asked for, and obtained two new evaluations to ensure a total of three evaluations on which to base my decision. Both reviewers clearly like the work and feel that it is of good quality and significant medical relevance. However, they both also find that some experiments need to be more rigorous, i.e. lack of dose response (for palbociclib treatments) and kinetics (for gH2AX) in the in vitro experiments, which are important to establish clinical relevance) and unclear relationship between the effects of the CDK inhibitor Palbociclib and CDK6 knockdown. Also unclear is the mechanistic basis of the effects of CDK6 KD.

Obviously, the criticisms raised by reviewer 2 from the first round of review (also pasted below for your convenience) remain and you will see there is some overlap with the new ones. One item of overlap is the weakness of the experimental findings from the in vivo data.

There is clearly here an overall consensus that the manuscript requires much attention, many controls, significant re-writing but all, including myself, agree that the potential interest of your findings is high. In conclusion, while publication of the paper cannot be considered at this stage, given the potential interest of your findings and after internal discussion, we have decided to give you the opportunity to address the above concerns.

We are thus prepared to consider a substantially revised submission, with the understanding that the Reviewers' concerns must be addressed with additional experimental data where appropriate to achieve substantial improvement of data quality, molecular analysis and experimental support for the main conclusions, and that acceptance of the manuscript will entail another round of review.

Since the required revision in this case appears to require a significant amount of time, additional work and experimentation, I would therefore understand if you chose to rather seek publication elsewhere at this stage. Should you do so and although we hope not, we would appreciate a message to this effect. Please note that it is EMBO Molecular Medicine policy to allow a single round of revision only and that, therefore, acceptance or rejection of the manuscript will depend on the completeness of your responses included in the next, final version of the manuscript.

As you know, EMBO Molecular Medicine has a "scooping protection" policy, whereby similar findings that are published by others during review or revision are not a criterion for rejection. However, I do ask you to get in touch with us after three months if you have not completed your revision, to update us on the status. Please also contact us as soon as possible if similar work is published elsewhere.

Please note that we now mandate that all corresponding authors list an ORCID digital identifier. You may do so through our web platform upon submission and the procedure takes <90 seconds to complete. We also encourage co-authors to supply an ORCID identifier, which will be linked to their name for unambiguous name identification.

I look forward to seeing a revised form of your manuscript in due course.

***** Reviewer's comments *****

Referee #3 (Comments on Novelty/Model System):

This is a sound paper that only requires some additional data points.

Referee #3 (Remarks):

In this manuscript the authors try to convince us that inhibition of CDK6 kinase activity enhances cell death in ovarian cancer by phosphorylating FOXO3 thereby regulating its stability. This is a nice and interesting paper that comes timely as CDK4/6 inhibitors are entering the clinics. There are only few treatment options for patients suffering from ovarian cancer which enhances the significance of that study. I have some questions:

The authors use a concentration of palbociclib that is somehow high and probably cannot be reached *in vivo*. The authors should repeat the experiments *in vitro* using the complete range of palbociclib dosages. Thereby they should also confirm their claim that palbociclib acts in a synthetic lethal manner in combination with platin-based drugs. As these are critical but easy experiments two different cellular systems should be used.

What is the kinetic of gamma-H2AX? First effects are seen after 5min. As the effects are not that convincing - could the authors perform kinetics - this may improve the outcome of the experiments and support their claims.

Can the authors comment why there are rather huge outliers in Figure 3G?

I am confused by the co-immunoprecipitation experiments depicted in Figure 4E. Why are there divergent outcomes after 3 and 6 hours? This stresses my point that better kinetic studies are required.

Can the authors ChIP CDK6 at the ATR promoter region?

Figure 6: The figure should be complemented using apoptosis staining and senescence stainings. Why do the authors suddenly switch to chromosomal condensation?

There are two good recent reviews that the authors may cite: Hydbring et al, 2016 and Tigan et al 2016.

Referee #4 (Comments on Novelty/Model System):

Technical quality is OK, the mechanisms and outcomes are relatively novel, and the work could have some medical impact.

Animal studies are medium quality and could have been substantially improved (ie. more models) to enhance the impact of the work.

Referee #4 (Remarks):

SUMMARY: The present manuscript evaluates the impact of CDK6 on the sensitivity of ovarian cancer cells to platinum-based therapy in preclinical models. Contrary to what may be expected based on prior studies, the knockdown of CDK6 is shown to potentiate the cytotoxic effects of carboplatin, and leads to a significant decrease in the apparent IC50. Modeling scheduling with the CDK4/6 inhibitor illustrates that providing the CDK4/6 inhibitor after treatment with carboplatin/cisplatin is necessary for the impact on sensitivity. The combination regimen appeared to have more activity in xenograft models. Analysis of mechanism suggests that CDK6 phosphorylates FOXO3, which in turn regulates ATR. Thus, with an absence of CDK6 there is ostensibly less capacity to mediate DNA-damage checkpoints. Lastly, it is shown using.

CRITIQUE: The study is interesting and would suggest a selective role for CDK6 in modulating the response to platinum based therapies. Overall the work is well performed, although there are some significant questions that remain poorly resolved as enumerated below:

1. It has been proposed that agents such as PD-0332991 compromise the cytotoxic activity of agents such as cisplatin by limiting cell cycle progression. There is little analysis in the manuscript of how CDK6 knockdown impacts on cell cycle, and/or the corresponding effect of PD-0332991 which should clearly arrest the cells. This concern is particularly relevant when considering that the phenotype ostensibly induced by CDK6 knockdown in pre-mature mitosis. The basis for this conclusion seems poorly supported.

2. How CDK6 has no impact on proliferation, but contributes to significantly increased senescence is unclear (Fig 1); particularly considering there is no effect on RB phosphorylation.

3. The data with PD-0332991 are inferred as mediating a CDK6-dependent effect. However, there is really little data that would support this conclusion. Since PD-0332991 inhibits proliferation in these cells the data in Figures 2C and in mice Figure 3 could have very little to do with the phenotype that appears to be specific to CDK6 knockdown. To support the authors' model, they would need to enforce the expression of FOXO3 in these xenograft and demonstrate that the effect of PD-0332991 is ameliorated. The overall efficacy data is also not particularly robust in the xenograft study.

4. The functional connection to ATR seem tenuous, and the data in figure 6 contradict other data relative to CDK6 regulating the levels of FOXO3.

5. The prognostic data with elevated levels of CDK6 could be due to a myriad of factors that are independent of the sensitivity to platinum-based therapy. Also given the proposed mechanism through which CDK6 regulates FOXO3 the correlation at the level of transcript seem contradictory?

Referee #2 (from the first round)

Platinum-resistance is a major issue in the treatment of HGEOC. This manuscript highlights several interesting findings seeking to elucidate novel drugs and therapeutic targets for the treatment of platinum resistant HGEOC. The authors show that CDK6 inhibition is synthetic lethal with platinum and platinum+ CDK6 inhibitor combination was effective in preventing the re-growth of platinum-resistant cells. Additionally, the authors propose that the effects of CDK6 inhibition are mediated through the regulation of FOXO3 and ATR. Though there are some interesting and novel findings in this manuscript there are several experiments that are required in order to strengthen the claims proposed by the authors.

- 1) For the initial shRNA screen, was there confirmation of knockdown for all 23 CDK's?
- 2) The authors mainly use the MDAH and OVCAR8 cell lines but the results are not consistently presented, ie, in Fig 1E MDAH shows CDK6 expression for just one day but OVCAR8 show the expression for 3 days, also in Fig1F the senescence assay was only performed in MDAH and not OVCAR8. It would be interesting to see whether the same effects are seen in OVCAR8 which actually seem to be more responsive to shCDK6 (Fig 1D).
- 3) Colony formation assays should be performed with the platinum+shCDK6 in order to assess the effects of long-term exposure to platinum as well as assessing apoptosis.
- 4) In order to confirm that CDK6 is a potent target in the platinum resistant setting, overexpression experiments should be performed in order to assess whether increased expression of CDK6 decreases sensitivity to platinum. Furthermore, if CDK6 is driving resistance its expression should be increased in platinum-resistant cells therefore the expression of CDK6 should be determined in the platinum-resistant clones generated from the MDAH cells.
- 5) In order to confirm that platinum + the CKD6 inhibitor (PD) is an effective combination therapy the authors should calculate the combination index or perform an isobologram analysis.
- 6) Though the authors mention that the effects of the PD compound are CDK6-dependent this can not be determined unless re-expression of CDK6 in the low expressing cells sensitize the cells to PD or direct inhibition of CDK6 using CRISPER or shRNA leads to PD resistance.
- 7) The in vivo studies using platinum + PD are promising but not dramatic and Fig 3B should be plotted as tumor volume not fold change over time in order to confirm that all tumors started treatment at a similar size. Furthermore, it is not clear whether Fig 3C is tumor volume at the end of the study, if so, the tumors were still very small when the study ended therefore it would be important to show what size the tumors were when they started treatment.
- 8) It is clear that FOXO3 and CDK6 bind but the half-life studies are not significant: in Fig. 5A it would be expected that the sh-CDK6 should start off with less FOXO3 at time point 0. Can the authors explain why we do not see a decrease at this time point?

9) Lastly, for the combination studies in the primary cell lines, there are clearly differences in response that cannot be attributed to the endogenous levels of CDK6, FOXO3 or ATR. Have the authors assessed the levels of these proteins post-treatment? The authors need to determine whether these cell lines behave the same way was MDAH and PD + platinum treatment results in degradation of CDK6, FOXO3 and/or ATR.

1st Revision - authors' response

20 June 2017

Referee #3 (Comments on Novelty/Model System):

This is a sound paper that only requires some additional data points.

Referee #3 (Remarks):

In this manuscript the authors try to convince us that inhibition of CDK6 kinase activity enhances cell death in ovarian cancer by phosphorylating FOXO3 thereby regulating its stability. This is a nice and interesting paper that comes timely as CDK4/6 inhibitors are entering the clinics. There are only few treatment options for patients suffering from ovarian cancer which enhances the significance of that study.

We really thank the Referee for His/Her general appreciation of our work

I have some questions:

1) The authors use a concentration of palbociclib that is somehow high and probably cannot be reached in vivo. The authors should repeat the experiments in vitro using the complete range of palbociclib dosages. Thereby they should also confirm their claim that palbociclib acts in a synthetic lethal manner in combination with platin-based drugs. As these are critical but easy experiments two different cellular systems should be used.

We thank the Referee for this suggestion. We have now calculated whether the PD-platinum combination has synergistic, additive, or antagonistic value, in MDAH, OVCAR8 and SKOV3ip cells, by applying the Chou-Talalay method with the Calcsyn software. We have calculated the combination index (CI) that is used to assess if two drugs have a synergistic (CI<1), additive (CI=1) or antagonistic effect (CI>1). The results, now included in the new **Fig 2C**, clearly demonstrate a strong synergism at all points considered and in all three cell lines.

2) What is the kinetic of gamma-H2AX? First effects are seen after 5min. As the effects are not that convincing - could the authors perform kinetics - this may improve the outcome of the experiments and support their claims.

As requested, we have repeated the kinetics for γ H2AX expression in the RB proficient MDAH and in the RB deficient OVCAR8 cells. The results, now shown in new **Fig 1I and Fig EV1D**, fully confirm that CDK6 knock down resulted in increased and more rapid appearance of DNA damage and increased apoptosis (evaluated looking at the levels of cleaved PARP1 and cleaved Caspase 3).

3) Can the authors comment why there are rather huge outliers in Figure 3G?

It is conceivable that the different number of Ki67 positive cells are due in part to the intrinsic variability of biological replicates and in part to the different size of tumors that were included in OCT for IF analyses. It is possible that in the largest masses analyzed (*i.e.* tumors treated with platinum alone), regions with different proliferation index coexist, resulting in a higher variability in Ki67 staining. However, in order to get more robust and consistent results, we have now confirmed our findings in a different *in vivo* model and using a different technique (*i.e.* Western Blot) (new **Fig 3D and Fig EV3**).

4) I am confused by the co-immunoprecipitation experiments depicted in Figure 4E. Why are there divergent outcomes after 3 and 6 hours? This stresses my point that better kinetic studies are required.

We agree with the Referee and have now performed a better kinetics for co-IP experiments, shown in new **Fig 4E**. The new data confirm that there is an increased association between FOXO3 and CDK6 that is both time- and dose-dependent. In fact using 7.5µg/ml of platinum the peak of association was observed at 16 hours of treatment, while using 15µg/ml the peak of association was observed between 3 and 9 hours of treatment.

5) *Can the authors ChIP CDK6 at the ATR promoter region?*

We tested if CDK6 directly binds the ATR promoter but we could not appreciate any significant occupancy of the ATR promoter by CDK6, in control or in PT-treated cells using the Chip assay (see new **Appendix Fig S3F**).

6) *Figure 6: The figure should be complemented using apoptosis staining and senescence staining.*
7) *Why do the authors suddenly switch to chromosomal condensation?*

We thank the Referee for highlighting this point, thus prompting us to better clarify the data presented in Figure 6.

Our experiments shown in Figure 1 and 2 clearly pointed to the fact that CDK6 silenced cells and treated with platinum or with platinum+PD underwent apoptosis. This was demonstrated at molecular level looking at cleaved PARP1 and cleaved caspase 3 expression (new **Fig 1I and Fig EV1D**). We confirmed the data using video time lapse microscopy (**Appendix Movies S1-S5**). Furthermore, using FACS analysis of the cell cycle progression, we observed that combination of platinum and CDK6 inhibition (using PD) resulted first in S phase block, then in G2/M of the cell cycle followed by increased apoptosis (sub G1 population in new **Fig 2H and Fig EV2J**).

We looked at the premature chromosome condensation (PCC) based on the notion that ATR inhibition during S phase could result PCC that then caused increased apoptosis. Thus we looked at PCC to explain how apoptosis was triggered by the inhibition of CDK6 and the concomitant platinum treatment knowing that in EOC cells platinum treatment induces an S phase block of the cell cycle. At the same manner we excluded the involvement of senescence in the observed phenotype using β-Gal assay in control and CDK6 silenced cells or in cells treated with PD in the presence or absence of platinum (new **Fig 1H, Fig EV2F and Appendix Fig S2D**). In all these experiments although a significant induction of senescence could be observed, we considered the absolute number of β-Gal positive cells to low to explain the massive cell death observed.

We better explained this points at page 13 in the results section and at page 17 in the discussion section.

8) *There are two good recent reviews that the authors may cite: Hydrbring et al, 2016 and Tigan et al 2016.*

We thank the Referee for this suggestion.

Referee #4 (Comments on Novelty/Model System):

Technical quality is OK, the mechanisms and outcomes are relatively novel, and the work could have some medical impact.

Animal studies are medium quality and could have been substantially improved (ie. more models) to enhance the impact of the work.

Referee #4 (Remarks):

SUMMARY: The present manuscript evaluates the impact of CDK6 on the sensitivity of ovarian cancer cells to platinum-based therapy in preclinical models. Contrary to what may be expected based on prior studies, the knockdown of CDK6 is shown to potentiate the cytotoxic effects of carboplatin, and leads to a significant decrease in the apparent IC50. Modeling scheduling with the CDK4/6 inhibitor illustrates that providing the CDK4/6 inhibitor after treatment with

carboplatin/cisplatin is necessary for the impact on sensitivity. The combination regimen appeared to have more activity in xenograft models. Analysis of mechanism suggests that CDK6 phosphorylates FOXO3, which in turn regulates ATR. Thus, with an absence of CDK6 there is ostensibly less capacity to mediate DNA-damage checkpoints. Lastly, it is shown using CRITIQUE: The study is interesting and would suggest a selective role for CDK6 in modulating the response to platinum based therapies. Overall the work is well performed, although there are some significant questions that remain poorly resolved as enumerated below:

We thank the Referee for His/Her general appreciation of our work .

1. It has been proposed that agents such as PD-0332991 compromise the cytotoxic activity of agents such as cisplatin by limiting cell cycle progression. There is little analysis in the manuscript of how CDK6 knockdown impacts on cell cycle, and/or the corresponding effect of PD-0332991 which should clearly arrest the cells. This concern is particularly relevant when considering that the phenotype ostensibly induced by CDK6 knockdown in pre-mature mitosis. The basis for this conclusion seems poorly supported.

We agree with the Referee and we have now better defined the role of CDK6 and PD in the control of cell cycle progression in EOC cells. Our experiments, using FACS analyses and growth curve experiment, clearly demonstrate that specific CDK6 knock down does not significantly impact on cell proliferation, in both RB-proficient and –deficient cells (new **Fig 1E and G Fig EV1C and Appendix Fig S2B**).

Similarly, we looked at the effects of PD-0332991 on cell proliferation and, as expected from the concomitant inhibition of CDK4 and CDK6, it arrested the cell cycle in the G1 phase. Accordingly, this RB-dependent effect was not observed in the RB-null OVCAR8 cells (new **Fig 2H and Fig EV2J**).

However, when administered after platinum, PD sustained the block in S phase that eventually resulted in increased cell death (sub G1 cells in **Fig 2H and Fig EV2J**). Accordingly, we noticed that the synergistic effects of platinum+PD could be observed only when PD was administered after platinum (**Fig 2B**). We have now better described these results at page 9 of the Results and at page 16 of the Discussion.

2. How CDK6 has no impact on proliferation, but contributes to significantly increased senescence is unclear (Fig 1); particularly considering those is no effect on RB phosphorylation.

We agree with the Referee that these data seem at first contrasting. It has to be taken into account that the β -Gal assay is a single cell assay while RB-phosphorylation was observed in Western Blot analyses. Since very few cells silenced for CDK6 became senescent (1-2 positive cells/field using a 40X objective, that means well below 1% of positive cells) it is likely that this very low, although significant, increase in senescence does not impact on total phospho-RB levels nor on the observed EOC cell death. To strengthen our results, we have now repeated the β -Gal assay in SKOV3 (RB-proficient) cells and verified that no difference in β -Gal positive cells between control and CDK6 silenced cells was present (new **Appendix Fig S2D**). Finally, we tested if PD alone or in combination with platinum increased the number of β -Gal positive cells. Again, we observed that, although significant, this increase was limited to a very small number of cells (1-3 positive cells/field using a 40X objective, new **Fig EV2I**).

3. The data with PD-0332991 are inferred as mediating a CDK6-dependent effect. However, there is really little data that would support this conclusion. Since PD-0332991 inhibits proliferation in these cells the data in Figures 2C and in mice Figure 3 could have very little to do with the phenotype that appears to be specific to CDK6 knockdown. To support the authors' model, they would need to enforce the expression of FOXO3 in these xenograft and demonstrate that the effect of PD-0332991 is ameliorated. The overall efficacy data is also not particularly robust in the xenograft study.

We agree with the Referee and we have tried to address these concerns in different ways, here schematically reported:

- 3a We have repeated the *in vivo* experiments using a different model (SKOV3-ip cells) and a longer period of treatment, confirming the efficacy of the combination treatment (new **Fig 3** and **Fig EV3**).
- 3b. We have better calculated the synergism between platinum and PD, using the Chou-Talalay method with the Calcsyn software (new **Fig 2C**)
- 3c. We have overexpressed CDK6 in OVCAR8 cells, demonstrating that high levels of CDK6 conferred a dose dependent resistance to platinum (new **Fig EV1E**).
- 3d. We have demonstrated that platinum resistant MDAH cells display higher levels of CDK6 and are more sensitive to the platinum+PD combination than parental MDAH cells (new **Fig 2D-G** and **Fig EV2G-H**).
- 3e. We have overexpressed FOXO3 WT and FOXO3 S325A (non phosphorylable by CDK6) in MDAH cells, demonstrating that only the WT protein reverted the increased cell death caused by platinum in CDK6 silenced cells (new **Fig 4D and 5G**).
- 3f. We have better defined the timing of CDK6-FOXO3 interaction (new **Fig 4E**).
- 3g. We have demonstrated that PD treatment prevents the platinum induced nuclear translocation of FOXO3 (new **Fig 5H**).
- 3g. We have demonstrated that PD treatment *in vivo* reduced FOXO3 expression level and its nuclear localization (new **Fig 5I**).

4. The functional connection to ATR seem tenuous, and the data in figure 6 contradict other data relative to CDK6 regulating the levels of FOXO3.

We have now presented clearer data sustaining the functional connection between CDK6-FOXO3 and ATR expression (new **Fig 6A**).

We also looked at the expression of ATR in MDAH cells treated with platinum with or without PD, showing that following PD treatment ATR expression decreases, overall supporting our hypothesis (new **Fig 6C**).

Similarly, the expression of ATR decreases in primary EOC culture treated with platinum and PD, confirming the relevance of this observation also in the human pathology (new **Fig 7C**). Altogether, we have now better characterized the model describing the role of CDK6 in controlling the expression and nuclear localization of FOXO3 in response to platinum.

5. The prognostic data with elevated levels of CDK6 could be due to a myriad of factors that are independent of the sensitivity to platinum-based therapy. Also given the proposed mechanism through which CDK6 regulates FOXO3 the correlation at the level of transcript seem contradictory?

We agree with the Referee that the prognostic data with elevated levels of CDK6 could be due to many factors other than platinum sensitivity. Yet, we would like to highlight that these data consistently go in the direction that we expected and that the same is not true for CDK4. This is intriguing since CDK4 shares with CDK6 most, if not all, of the RB-dependent cell cycle related functions.

Moreover, we have reinforced our data by including new analyses showing an increased expression and increased correlation between CDK6 and FOXO3 or ATR proteins in samples from platinum treated patients (**Appendix Fig S4A-B**). Finally, using multivariate analyses we demonstrate that the prognostic role of CDK6 mRNA expression, in the Tothill dataset, is independent from the stage and the optimal debulking of primary tumor that in EOC patients represent the two most significant variables.

Overall, these observations support the possibility that CDK6 could be used as marker of poor prognosis in EOC patients and to select patients that may benefit from a combination treatment based on platinum + PD.

Referee #2 (from the first round)

Platinum-resistance is a major issue in the treatment of HGEOC. This manuscript highlights several interesting findings seeking to elucidate novel drugs and therapeutic targets for the treatment of platinum resistant HGEOC. The authors show that CDK6 inhibition is synthetic lethal with platinum and platinum+ CDK6 inhibitor combination was effective in preventing the re-growth of platinum-resistant cells. Additionally, the authors propose that the effects of CDK6 inhibition are mediated through the regulation of FOXO3 and ATR. Though there are some interesting and novel findings in this manuscript there are several experiments that are required in order to strengthen the claims proposed by the authors.

We thank the Referee for finding of potential interest our work and for His/Her suggestions to improve the strength of our results.

1) For the initial shRNA screen, was there confirmation of knockdown for all 23 CDK's?

To address this point, we have first checked the expression level of the 23 CDKs in MDAH cells. We found that some of them were expressed at very low/undetectable level (*i.e.* a concentration \leq 0.00015 amol/ng of total RNA that corresponds to a normalized expression, using Pol2A = 0.0015 amol/ng, lower than 0.1).

Then, we tested the activity of the single shRNAs only for those CDKs that had a detectable mRNA expression (\geq 0.00015 amol/ng). The new data, now shown in the new **Appendix Fig S1C**, demonstrate that in all cases at least one shRNA significantly reduced the expression of the targeted CDK. These results were confirmed by Western Blot in the case of CDK4 (new **Appendix Fig S1D**).

2) The authors mainly use the MDAH and OVCAR8 cell lines but the results are not consistently presented, ie, in Fig 1E MDAH shows CDK6 expression for just one day but OVCAR8 show the expression for 3 days, also in Fig1F the senescence assay was only performed in MDAH and not OVCAR8. It would be interesting to see whether the same effects are seen in OVCAR8 which actually seem to be more responsive to shCDK6 (Fig 1D).

We agree with the Referee and we have now recapitulated all the experiments *in vitro* and *in vivo* using OVCAR8 as a model of RB null cells and/or SKOV3ip as a model of platinum resistant cells (see new **Fig 1, Fig EV1 and EV4 and Appendix Fig S2**). Thus, all experiments are now performed in at least two models, as requested.

3) Colony formation assays should be performed with the platinum+shCDK6 in order to assess the effects of long-term exposure to platinum as well as assessing apoptosis.

As suggested we have performed a colony formation assay in cells silenced for CDK6 and treated or not with platinum. The data are now shown in the new **Fig 1D and Appendix Fig S2B**.

4) In order to confirm that CDK6 is a potent target in the platinum resistant setting, overexpression experiments should be performed in order to assess whether increased expression of CDK6 decreases sensitivity to platinum. Furthermore, if CDK6 is driving resistance its expression should be increased in platinum-resistant cells therefore the expression of CDK6 should be determined in the platinum-resistant clones generated from the MDAH cells.

We agree with the Referee and we now show that overexpression of CDK6 in OVCAR8 cells increases their resistance to platinum in a dose dependent manner (new **Fig EV1E**). At low platinum concentration this effect was better observed using a constitutively active CDK6 mutant (new **Fig 2A**). We checked the expression of CDK6 in platinum -resistant MDAH cells showing that these cells had higher CDK6 expression, both at protein and mRNA level (new **Fig EV2G-H**)

5) In order to confirm that platinum + the CKD6 inhibitor (PD) is an effective combination therapy the authors should calculate the combination index or perform an isobologram analysis.

We thank the Referee for this suggestion.

We have performed a dose response analysis for PD treatment, alone and in combination with platinum, to precisely calculate whether this combination is synergistic, additive, or antagonistic. We applied the Chou-Talalay method using the Calcsyn software to calculate the combination index (CI) and assessed if two drugs have a synergistic (CI<1), additive (CI=1) or antagonistic effect (CI>1). The results now included in the new **Fig 2C** demonstrate a moderate-strong synergism, at all concentrations considered and in all three cell lines tested (*i.e.* MDAH, OVCAR8 and SKOV3).

6) Though the authors mention that the effects of the PD compound are CDK6-dependent this can not be determined unless re-expression of CDK6 in the low expressing cells sensitize the cells to PD or direct inhibition of CDK6 using CRISPER or shRNA leads to PD resistance.

We have now demonstrated that PD had no effect on platinum sensitivity in CDK6 silenced cells, in cells expressing very low levels of CDK6 while it is more active in PT-resistant MDAH cells that express high levels of endogenous CDK6 when compared to parental MDAH cells (new **Fig 2 and Fig EV2**).

We also repeatedly tried to overexpress CDK6 in OVSAHO and KURAMOCHI cells but, unfortunately, these cells resulted very refractory to transfection and we were not able to obtain a significant population of CDK6 overexpressing cells.

7) The in vivo studies using platinum + PD are promising but not dramatic and Fig 3B should be plotted as tumor volume not fold change over time in order to confirm that all tumors started treatment at a similar size. Furthermore, it is not clear whether Fig 3C is tumor volume at the end of the study, if so, the tumors were still very small when the study ended therefore it would be important to show what size the tumors were when they started treatment.

We absolutely agree with the Referee on the necessity to perform new *in vivo* experiments. In this case, as suggested, we have tested a different model (*i.e.* SKOV3-ip cells) that is considered a model of platinum resistance and we have performed two different experiments. In the first case, we have treated small tumors (~ 50mm³) with vehicle, PD, platinum or their combination and monitored the animals for three weeks after the start of treatments.

As a second approach, we have treated larger tumors (~ 150mm³) with vehicle or platinum+PD, to verify if the combination treatment could have a therapeutic impact also in larger masses and monitored the animals for two weeks. The new data, shown in new **Fig 3 and Fig EV3**, demonstrated that in both cases the platinum+PD treatment significantly reduced tumor growth, cell proliferation (Ki67 expression) and increased DNA damage (γ H2X phosphorylation), further supporting the possibility that this combination has therapeutic potential.

8) It is clear that FOXO3 and CDK6 bind but the half-life studies are not significant: in Fig. 5A it would be expected that the sh-CDK6 should start off with less FOXO3 at time point 0. Can the authors explain why we do not see a decrease at this time point?

We regret for the confusion and the inaccuracies present in the previous version of the manuscript. We have now better explained this point in **Fig 5A** and in the accompanying text. In the absence of platinum (vehicle treated cells) we did not observe differences in FOXO3 half-life. On the contrary, after platinum treatment (CDDP) due to the schema of treatments and the kinetic of CDK6/FOXO3 association (see new **Fig 4E**), we observed major differences in the kinetics of degradation. After platinum treatment at Time 0, FOXO3 levels normalized over tubulin (n=3 experiments) were 13.4±2.8 in control and 11.4±1.9 in CDK6 silenced cells (p=0.0006). These values were considered the 100% amount of FOXO3 for control and CDK6 silenced cells. After 2 and 4 hours of CHX treatments the remaining levels of FOXO3 were 100.7±1.6 and 102.2±3.5 in control and 85.5±2.5 and 67.1±2.3 in CDK6 silenced cells (p=0.0006 and p=0.0002, respectively).

The data are presented as percentage respect to time 0 in each condition for simplicity and the significant differences are now reported in the new **Fig 5A**.

9) Lastly, for the combination studies in the primary cell lines, there are clearly differences in response that cannot be attributed to the endogenous levels of CDK6, FOXO3 or ATR. Have the

authors assessed the levels of these proteins post-treatment? The authors need to determine whether these cell lines behave the same way was MDAH and PD + platinum treatment results in degradation of CDK6, FOXO3 and/or ATR.

We agree with the Referee on the importance to confirm our results in primary EOC culture. The new collected data, now shown in the new **Fig 7C**, demonstrate that platinum treatment in primary EOC cultures induces an increase in ATR expression and that this increase is not observed in cells treated with platinum + PD, confirming that when primary EOC cells responded to the combination treatment (i.e. #66, #77 and #91a) the induction of ATR expression by platinum was inhibited.

3rd Editorial Decision

07 July 2017

Thank you for the submission of your manuscript to EMBO Molecular Medicine. We are sorry that it has so long to get back to you on your manuscript.

We experienced some difficulties in securing the evaluations in a timely fashion. Further to this, we did not succeed in obtaining an evaluation from reviewer 4.

As you will see reviewers 2 and 3 are now satisfied with your manuscript. As for your responses to reviewer 4, we evaluated them at the editorial level to be satisfactory.

I am thus prepared to accept your manuscript for publication pending the following editorial amendments:

- 1) As per our Author Guidelines, the description of all reported data that includes statistical testing must state the name of the statistical test used to generate error bars and P values, the number (n) of independent experiments underlying each data point (not replicate measures of one sample), and the actual P value for each test (not merely 'significant' or 'P < 0.05'). Based on practical and/or esthetical considerations on the figures, you may opt to provide the P values in a dedicated table in the appendix file. Should you decide to choose this option, please make sure to reference the table with correct callouts and in appropriate sections of the manuscript, e.g. figure legends.
- 2) The manuscript must include a statement in the Materials and Methods identifying the institutional and/or licensing committee approving the experiments, including any relevant details (like how many animals were used, of which gender, at what age, which strains, if genetically modified, on which background, housing details, etc). I note that you have not provided all this information in your manuscript. We encourage authors to follow the ARRIVE guidelines for reporting studies involving animals. Please see the EQUATOR website for details: <http://www.equator-network.org/reporting-guidelines/improving-bioscience-research-reporting-the-arrive-guidelines-for-reporting-animal-research/>. Please make sure that ALL the above details are reported in the main manuscript as well as the checklist
- 3) In Fig. 4A please make sure that the FOXO3 strip is contained within the box
- 4) Please provide a size bar for Fig. 6h
- 5) Please reduce excessive contrasting in the following figures: 4d, 4e, 6a, 6c, EV1b, EV2a, EV3k, EV4d, EV4e, EV5f.
- 6) The Appendix TOC requires the indication of page numbers and should be provided as a PDF file. Also, please remove the movie legends from the Appendix; they should instead be included in the zipped movie files. Please rename movies as Movie EV1 etc. not Appendix Movie S1 etc. (with corrected manuscript callouts).
- 7) In general, the quality of the appendix figures is quite low. Please try to improve them.
- 8) Thank you for providing the source data files. However please resolve the following inconsistencies:
Fig4E: CDK6 7.5 ug/ml and 15 ug/ml appear to be mismatched in the source data (SD) file vs. actual figure;

Fig.7B: I find it difficult to match the cyclin D1 strip in SD vs. figure;
Fig.7E: unclear SD mapping;
Fig.EV1B: I could not match the CDK strip to the corresponding SD;
Fig.EV4E SD: probably the first strip should be labelled FOXO3?

Please submit your revised manuscript within two weeks. I look forward to seeing a revised form of your manuscript as soon as possible.

***** Reviewer's comments *****

Referee #2 (Remarks):

The authors have addressed all my concerns and the manuscript is greatly improved.

Referee #3 (Remarks):

My comments have been addressed.

2nd Revision - authors' response

10 July 2017

Authors made requested editorial changes.

Corresponding Author Name: Gustavo Baldassarre

Manuscript Number: EMM-2016-07012-V3